# Understanding others' distal goals from proximal communicative actions

**Martin Dockendorff** [1]\*, **Laura Schmitz**[2], **Cordula Vesper**[3,4], **Günther Knoblich**[1]

**1** Department of Cognitive Science, Central European University, Vienna, Austria, **2** Department of Neurology, University Medical Center Hamburg-Eppendorf, Hamburg, Germany, **3** Department of Linguistics, Cognitive Science, and Semiotics, Aarhus University, Aarhus, Denmark, **4** Interacting Minds Centre, Aarhus University, Aarhus, Denmark

\* Dockendorff_martin@phd.ceu.edu

**Data Availability Statement:** All relevant data are within the manuscript and its Supporting information files.

**Funding:** This research was supported by the European Research Council under the European

## Abstract

Many social interactions require individuals to coordinate their actions and to inform each other about their goals. Often these goals concern an immediate (i.e., proximal) action, as when people give each other a brief handshake, but they sometimes also refer to a future (i.e. distal) action, as when football players perform a passing sequence. The present study investigates whether observers can derive information about such distal goals by relying on kinematic modulations of an actor's instrumental actions. In Experiment 1 participants were presented with animations of a box being moved at different velocities towards an apparent endpoint. The distal goal, however, was for the object to be moved past this endpoint, to one of two occluded target locations. Participants then selected the location which they considered the likely distal goal of the action. As predicted, participants were able to detect differences in movement velocity and, based on these differences, systematically mapped the movements to the two distal goal locations. Adding a distal goal led to more variation in the way participants mapped the observed movements onto different target locations. The results of Experiments 2 and 3 indicated that this cannot be explained by difficulties in perceptual discrimination. Rather, the increased variability likely reflects differences in interpreting the underlying connection between proximal communicative actions and distal goals. The present findings extend previous research on sensorimotor communication by demonstrating that communicative action modulations are not restricted to predicting proximal goals but can also be used to infer more distal goals.

## Introduction

People engage in a variety of complex social interactions that require temporal and spatial coordination of their individual actions, ranging from carrying a sofa with someone to performing a musical duet [1, 2]. In order to achieve such a feat, interaction partners often predict each other's actions by relying on behavioral cues from which they can derive useful anticipatory information about what their partners are about to do [3]. These cues, in turn, facilitate interpersonal coordination and the achievement of joint goals. One type of cue, which has received particular attention during recent years, consists of communicative modulations of

Union's Seventh Framework Program (FP7/2007-2013) / ERC grant agreement n° [609819], 961 SOMICS. The funders had no role in the study design, data collection and analysis, decision to publish, or preparation of the manuscript.

**Competing interests:** The authors have declared that no competing interests exist.

instrumental movements. For instance, two pianists playing a duet might lift their fingers higher, and by doing so, inform each other about the exact timing of their actions [4]. Similarly, when carrying a sofa together, the person who is walking forward might exaggerate an upward movement with the sofa to inform the person who is walking backward about the upcoming staircase [5]. By exaggerating their movements in this manner, interaction partners can fulfill two goals simultaneously: an instrumental goal, such as playing a piano piece or moving a sofa, and a communicative goal of informing an interaction partner about one's goals and intentions. This general capacity to provide anticipatory information about one's goals and intentions by means of communicative modulations of instrumental movements has been termed *sensorimotor communication* (SMC) [6].

A growing body of research has investigated SMC in experimental tasks in which two participants coordinate their actions to achieve a joint goal while the information relevant for attaining this goal was allocated asymmetrically between them (for a review, see [7]). "Leader" participants with full task information have been shown to spontaneously modulate certain kinematic features of their goal-directed movements, such as grip aperture [8], movement direction [9], movement amplitude [10, 11], and velocity [12, 13], to make their actions more informative, and hence more predictable for "Followers" participants who only have incomplete task information [13].

In order for SMC to be an effective form of communication, Followers need to be able to perceive the kinematic modulations in Leaders' goal-directed movements. Growing evidence indicates that observers can indeed perceive such modulations (e.g., [14–18]), and that they can understand them as conveying specific information about the leader's proximal (i.e. immediate) goals, such as reaching for a particular object [9] or aiming towards one of several target locations [10].

What allows for such sensitivity towards others' actions is the fact that observers use their own motor systems to predict others' unfolding actions and goals by generating internal simulations of the observed movement (e.g., forward models, see [19–21]). The predictive nature of such internal simulations enables the observer to revise and update her expectations about the actor's goals in a timely fashion, particularly in cases where the observed movements deviate from their most efficient performance, as is the case in SMC [6, 22]. These latter cases where actors deviate from efficient performance have been argued to constitute a proper form of communication to the extent that they enable observers to both derive useful anticipatory information about an actor's goals and to interpret and disambiguate between different, sometimes competing, goals [7]. Accordingly, we regard actions as communicative when they have the potential to facilitate an observer's prediction of another actor's upcoming goals. Note that by adopting this broad definition of communication, we focus on the receiver end of the interaction, i.e. how the receiver reacts to such communicative actions. In contrast, other perspectives on communication tend to highlight the role that the sender plays in producing communicative actions and they often require mutual awareness of communicative intentions in both producing and understanding these actions [23]. The broad definition on which we base our work is consistent with previous findings in sensorimotor communication (SMC) showing that, by interpreting a particular kinematic modulation as conveying specific information about an actor's goal, observers can adapt their own behavior in ways that facilitate interpersonal coordination and the achievement of a joint goal [8, 12, 13].

What is less clear from previous research is whether the use of SMC is restricted to facilitating predictions of immediate proximal goals or whether individuals can also interpret movement modulations that encode information about their partner's upcoming *distal goals*, i.e., goals that go beyond the observed action and thus are only attained after the achievement of a more proximal (sub-) goal first. The aim of the present study was to extend previous research

on SMC by focusing on how observers interpret communicative modulations of instrumental actions that convey information not only about proximal, but also about distal goals.

### From proximal to distal goals

To illustrate the difference between proximal and distal goals, consider a situation where a football player (Player A) recovers the ball on her side of the field and prepares a quick counterattack. Two of her teammates (Players B and C) start running along the flanks towards the opposite goal, ready to receive the ball. At this point, Player A could simply pass the ball to either of her teammates, thereby fulfilling her proximal goal. An alternative would be for Player A to continue dribbling the ball up to the midline, and to only then pass it on to one of her teammates. In this latter situation, the dribbling of the ball has become the more proximal goal, while the passing of the ball is now the more distal goal, since it follows temporally and is mediated by the prior achievement of a more proximal goal.

Although proximal and distal goals are separated in time, there is now strong evidence showing that distal goals can affect the kinematics of early components of proximal actions. For example, when individuals perform reach-to-grasp movements towards an object, different distal goals (e.g., throwing the object into a large box or placing it in a well) differentially affect the velocity of the early transport phase of the movement [24]. Relatedly, when participants perform two-step action sequences, the specific constraints imposed on the second action component (e.g., pouring from a bottle or throwing it) can influence the kinematics of the first component (e.g., grasping the bottle) [17, 25–27]. These findings can be interpreted in terms of a more general binding procedure that links both motor and perceptual features of a distal goal when organizing multiple movement segments within a "common event file" [28]. As a consequence of this binding, the activation of relevant perceptual features of a distal goal can lead to the concurrent activation of the appropriate motor program that is normally used to achieve that goal ([29], also see [30] for a similar argument, but supported by neurological evidence).

Similar "backpropagation effects" have been reported in social tasks, where distal social goals (e.g., passing an object to another person or placing it in front of her) have been shown to affect the kinematics of early action components (e.g., reaching towards the object, [14, 31, 32]). With respect to the perception and interpretation of these movements, recent findings suggest that observers can extract and use these early kinematic cues to discriminate between actions performed with different social intentions [15, 33] or to predict the outcome of a ballistic movement, such as when someone is throwing a ball [34].

The question that these studies leave open is whether such early effects of distal goals on the kinematics of proximal movements could also be used communicatively, i.e., in cases where agents intentionally make their distal goals easier to predict. To illustrate this idea, consider again our football example from above and assume that Player A wishes to inform her teammates that she will pass the ball to Player C, who is already much further down the field than Player B. To make this intention explicit to her teammates, Player A visibly increases her movement speed, thereby demonstrating that she is preparing for a long, powerful pass to Player C (rather than a short pass to Player B). By increasing her dribbling speed, Player A modifies the kinematics of an early action aimed at achieving a proximal goal in a way that provides useful information about her future distal goal to her two teammates. Importantly, the two teammates can use these early kinematic cues to predict and disambiguate between possible distal goals, and adapt their behavior accordingly, e.g., Player C can prepare to receive the ball, whereas Player B can try to draw the attention of the opposite's team defenders away from the passing sequence.

A recent computational account of early intention recognition of sequential actions formalizes the idea that observers can disambiguate between an observed agent's distal goals early, but only when the agent *co-articulates* the two movement primitives within the sequence [35]. Co-articulation, in this context, means that the agent alters the execution of an earlier proximal movement (e.g., reaching and grasping a bottle) in order to satisfy the specific constraints posed by the achievement of an upcoming, more distal goal (e.g., pouring from the bottle or simply moving it). Through a series of computational simulations as a proof-of-concept, Donnarumma and colleagues showed that when two sequential proximal movements are co-articulated, the kinematic features of the first movement are sufficient for an observer agent to correctly identify and disambiguate the distal goal. Importantly, their proposal also put forward the possibility that co-articulation might be used by actors strategically, as a way of helping an observer understand their distal goals (see the Appendix of [35]). Here, we address this possibility empirically by drawing on a) previous research showing that distal goals can affect early action components (e.g., [27]) and b) computational simulations suggesting the possibility to use early kinematic cues to disambiguate between distal goals [35].

## Motor iconicity in SMC

Although the research reviewed above suggests that observers might be able to use early action components to predict an upcoming distal goal, it leaves open the question of how communicative modulations of those same actions might be interpreted with respect to different distal goals. As highlighted earlier, movement deviations not only make proximal goals easier to predict, but also allow observers to disambiguate between potential action alternatives. For example, previous studies have shown that specific kinematic features of a Leader's movement, such as movement duration, height or direction, can be used to disambiguate between target locations that differ in terms of distance (near or far; [13]), height (upper or lower; [12]) or location (left or right; [9]). Such systematic mappings between kinematic features of movements and proximal goals have recently been described as "iconic" [13], because the relation between them involves some form of similarity which can be easily identified by both senders and receivers [36]. In the case of SMC, the relevant similarity is established between a particular movement used by the sender and the *most likely* goal that such movement achieves during natural performance. For example, a *higher* movement trajectory communicates a *higher* final grasp location [12]—because higher grasping movements are naturally performed with higher movement amplitude (also see [37]). A similar observation regarding the similarity between communicative movements and their goals has been made by researchers in sign language, who have argued that verbs in American Sign Language (ASL) used to refer to the manipulation of a tool (i.e., so-called handling classifier verbs such as BRUSH-HAIR or BOUNCE--BALL) are represented "motor-iconically" with a handshape that depicts how a person grasps and manipulates the tool, as well as the movements typically performed with it (e.g., brushing one's hair or bouncing a ball) [38]. Similarly to the case of SMC, the particular handshape and movements used in ASL to represent a tool correspond to the hand movements and goals that one would normally achieve while manipulating it. In line with these observations, we will henceforth refer to the relation between movements and their most likely goals as "motor-iconic".

In the case of distal goals, the motor-iconic relation between movements and goals can be captured by looking at the specific ways in which people perform an action when this same action is directed at a proximal goal. For example, when people perform unconstrained aiming movements towards far proximal targets they use higher peak velocity compared to near proximal targets [39]. If this relation is invoked in someone who observes such movements in order

to predict an agent's distal goal, then it can equally be considered as motor-iconic. In other words, the motor-iconic relation underlying the observation of movements and their distal goals is grounded on an understanding of that same movement were it to be directed towards a proximal goal. (Note that this way of defining iconicity departs in some ways from more standard definitions in semiotics and linguistics that focus on the relation between the form of a sign and a referent in the world [40]. Instead, motor-iconicity focuses on the relation between a particular form (e.g., a movement or a gesture) and the contents of a motor representation in the mind of the speaker or signer (e.g., a goal) [41, 42]).

Taken together, these considerations lead to the specific empirical question of whether observers map proximal communicative modulations of goal-directed movements onto proximal and distal goals in a motor-iconic manner, i.e., in a manner that preserves the link between the kinematic features of the proximal movement and its most likely proximal or distal goal.

## The present study

In a computer-based online experiment, participants observed animations of a box being moved at different velocities along a horizontal line from a start location towards a designated movement endpoint. Due to a partial occlusion of the visual scene, participants were not able to observe how the box actually reached the target location. They could therefore only rely on features of the observable proximal part of the movement to determine the likely final location of the box. After observing the animated movement, participants were asked to select the target location which they considered the likely proximal or distal goal of the action (Fig 1). In contrast to previous studies, in which participants observed movements in two- or three-dimensional space, participants in the present study observed animations of one-dimensional sliding

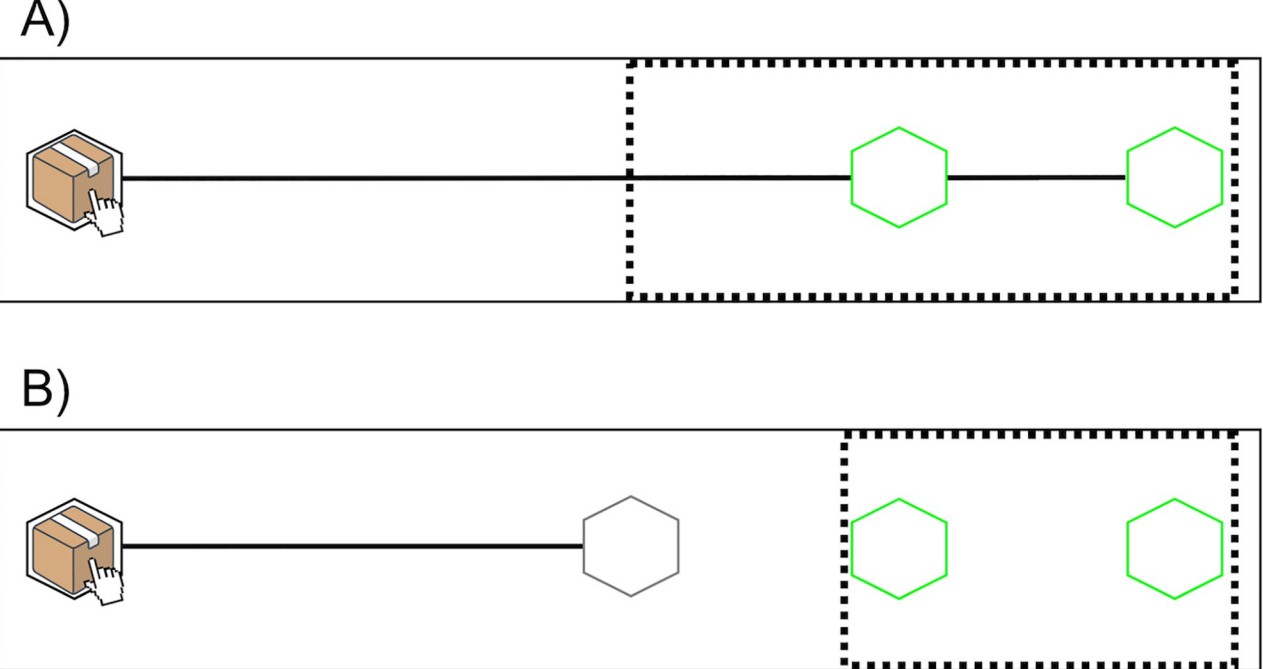

**Fig 1. Experimental layout used in Experiments 1–3.** Experimental layout in the (A) Proximal goal condition and the (B) Distal goal condition. The black dotted line represents the outline of the occluded area during trials, where the (near and far) target locations are displayed in light green.

movements in order to single out the role of temporal movement parameters (e.g. velocity and duration) for extracting information about the target locations.

To find out whether observers extract information about distal goals from early kinematic modulations, we manipulated whether the animated movements that participants saw achieved a proximal goal only (i.e., sliding a box to one of the two target locations) (Fig 1A: "Proximal goal" condition) or achieved a distal goal (i.e., delivering the box to one of the target locations) by means of achieving a proximal goal first (i.e., sliding the box towards a middle target) (Fig 1B: "Distal goal" condition). In this latter condition, the achievement of the distal goal was made possible by having the box disappear from the display and reappear within one of the target locations. The purpose of this "teleportation" was to introduce a visible spatial and temporal separation between the proximal movement and the distal goal. Such separation prevented participants from simply extrapolating the proximal sliding movement towards the distal goal since the movement endpoint was now shifted away from the target locations towards the middle of the display. Finally, this separation also enabled us to keep the Proximal and Distal goal conditions as similar as possible as both contained only a single sliding movement that participants needed to interpret.

Drawing on previous research on SMC, we had three central predictions: First, we predicted that participants would be able to detect differences in the velocity of the observed movements, and that this detection would allow them to consistently map these movements to one of the two potential goals (i.e., target locations). Second, we predicted that the stronger the communicative modulation of velocity, the easier participants' decision should be, resulting in higher consistency of their mappings. Third, based on the lawful relation between movement velocity and distance of natural movements, where farther target locations are reached with higher peak velocities in unconstrained aiming movements [39], we expected participants to map faster movements onto the far target location and slower movements onto the near target location. This mapping would represent what we call a motor-iconic relation, as it preserves the underlying link between observed movements and their most likely goals.

## Experiment 1: Interpreting velocity modulations

The aim of the first experiment was to establish the experimental paradigm by testing our main predictions regarding the role of distal goals in interpreting modulations in the velocity of proximal movements.

### Methods

**Participants.**   We recruited 50 participants (16 women; Age: *M* = 29.6 years; *SD* = 9.9 years), 25 per condition, through the online testing platform Testable (https://www.testable.org/). Sample size was determined using the Superpower statistical package [43] on R Studio [44]. We aimed at obtaining a medium effect size (.4) and high statistical power (>.8) based on a series of well-established findings showing that participants can detect subtle kinematic cues to predict other agents' goals (e.g. [14, 17]).

Participants were all proficient English speakers, and were paid 1.5£ for an estimated study completion time of 10 minutes. All participants gave prior written informed consent in accordance with the United Ethical Review Committee for Research in Psychology (EPKEB). This design and analysis of this study was pre-registered at https://osf.io/2qkn3. All data and materials are available on OSF, at https://osf.io/pv74b.

**Stimuli.**   The basic layout for each experimental condition of Experiment 1 is shown in Fig 1. In both conditions, participants saw a stationary box with a mouse cursor attached to it. The box and cursor were displayed within a black hexagonal location on the left side of the

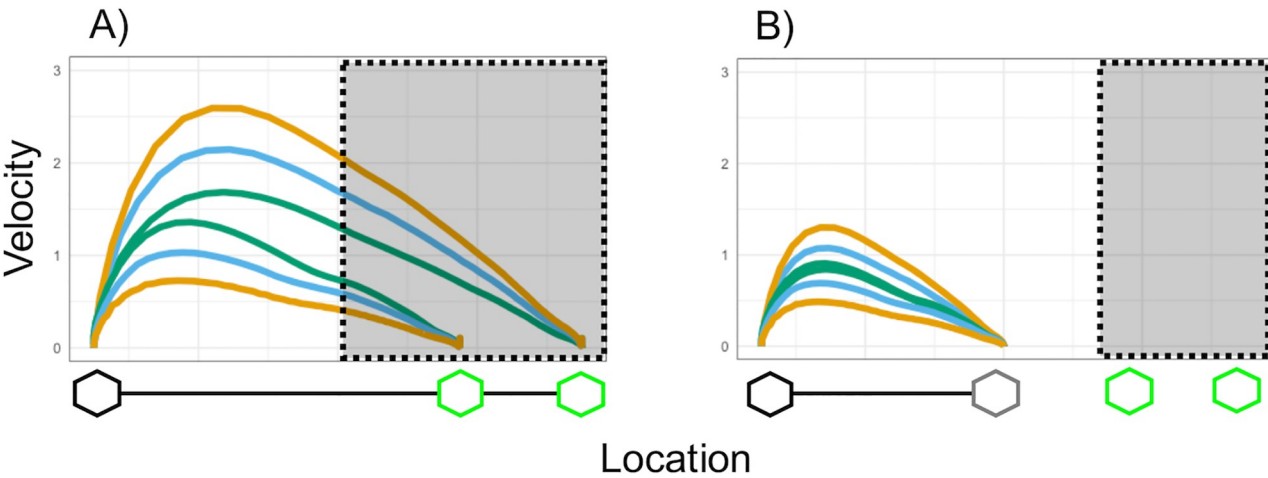

**Fig 2. Velocity profiles of sliding movements.** Velocity profiles of sliding movements used in the (A) Proximal goal condition and the (B) Distal goal condition. Normal movements are colored in green, Exaggerated movements in blue, and Very exaggerated ones in yellow. The dotted areas represent the occluded areas in each condition during experimental trials.

screen. During familiarization, participants also saw two green hexagonal target locations on the right side of the screen. During trials, these two target locations were covered with a rectangular black occluder.

A black horizontal line, along which the box moved during the trials, connected the initial location to the green target locations on the right side of the screen (Proximal goal, Fig 1A), or to a grey hexagonal area in the middle of the screen (Distal goal, Fig 1B).

*Movement animations.* The animations of the box movements were created from averaged actual mouse movements recorded with PsychoPy [45] in a setup identical to the one shown in Fig 1A (Proximal goal condition). Movements to near and far targets were averaged separately, thus obtaining two "prototypical" natural movements, one for each target location (henceforth "Normal near" and "Normal far" movements). Based on the two Normal movements, exaggerated movements were generated by, first, identifying the peak velocities for each (i.e., Normal near and Normal far) movement. Then, we rescaled both of these movements such that the peak velocity was either one or two standard deviations below the peak velocity of the Normal near movement, or one or two standard deviations above the peak velocity of the Normal far movement. This procedure led to overall six different movements (Fig 2A): two "Normal" ones (i.e., Normal near and Normal far), two "Exaggerated" ones (i.e., Slow and Fast) and two "Very exaggerated" ones (i.e., Very slow and Very fast).

To create the movement animations in the Distal goal condition, these six movements were further reshaped so that their endpoints would all converge towards the middle of the screen (Fig 2B). Critically, this procedure retained most kinematic features of the original movements (e.g., bell-shaped velocity profiles) but eliminated the differences in movement distance such that all movements now had the same endpoint in the middle of the screen. Further details on how movements were recorded and averaged are provided as in S1 Appendix, including details on how the rescaling affected the movements in the Distal goal condition.

**Design.** The experiment consisted of a mixed factorial design, with one between-subject variable (goal type) and one within-subject variable (degree of exaggeration). The between-subject variable manipulated whether the sliding movement achieved a proximal or distal goal. Specifically, the proximal goal consisted in simply reaching one of the two target locations (i.e., green targets) in the Proximal goal condition (Figs 1A and 2A). In the Distal goal condition,

the endpoint of the sliding movement was temporally and spatially separated from the final target locations to an intermediate target located in the middle of the screen. This meant that the goal of reaching one of the two target locations was only achieved by moving to this intermediate target first (Figs 1B and 2B). As a consequence, reaching one of the two actual target locations became the distal goal in this condition, while moving towards the middle target became the proximal one. The within-subject variable manipulated whether and to which degree the animated movements were exaggerated in terms of peak velocity (i.e., Normal (i.e., no exaggeration), Exaggerated, Very exaggerated).

## Procedure

**Familiarization and instructions.** After being randomly assigned to one of the goal type conditions, participants were familiarized with the task. They were first presented with the complete task layout (as illustrated in Fig 1), but without the occluder covering the green target locations. Participants in both conditions then saw two successive Normal movements of the box, one to the near target, the other to the far target (order counterbalanced across participants) (see Fig 2, Normal movements in green). In the Proximal goal condition, participants saw the box moving at a Normal velocity to one of the two green target locations (see Fig 2A). In the Distal goal condition, after participants had observed the box moving at Normal velocity to the middle of the screen, the box disappeared for approximately 500 ms and then reappeared in one of the two green target locations (note that in this condition the two Normal movements had overlapping velocity profiles, see Fig 2B and S1 Appendix for further details). After seeing these two movements, participants in both conditions were asked to select the target location where they saw the box had moved by pressing the "n" key (for near) or "f" key (for far).

Next, a black occluder covered the target locations and participants were told that during the actual experiment, they would be presented with another participant's previously recorded movements. Importantly, they were informed that this previous participant had produced the movements "in ways that would help others guess to which green target location he/she was moving the box". This information was provided in order to make it explicit to participants that the movements they were about to see were communicative, that is, that they contained useful information about the previous participant's goals.

**Experiment.** Participants performed 36 experimental trials, divided into six blocks. In each trial, they were presented with an animation of the box sliding along the black line to either the occluded target locations in the Proximal goal condition, or towards the middle of the screen in the Distal goal condition. In this latter condition, once the box reached the middle of the screen, it remained stationary for a few moments, and then disappeared from the display. Participants in both conditions were then prompted to answer to which location they thought the box had been delivered. A trial was completed when participants pressed one of the two assigned keys ("n" or "f"), corresponding to either the "near" or "far" target locations. Each block contained all six degrees of exaggeration, presented in random order. Participants did not receive feedback about their performance at any point. At the end of the experiment, participants were asked to fill out a short questionnaire about their experience with the task.

## Results

**Data preparation.** We categorized participants' responses as *Iconic* or *Non-Iconic* mappings. Iconic mappings refer to trials where participants pressed the "n" key in response to movements with lower peak velocity and the "f" key in response to movements with higher peak velocity. Note that Iconic mappings correspond to the "motor-iconic" relation, since they

preserve the relation between movement velocity and most likely goal. Non-Iconic mappings, on the other hand, refer to those trials where participants reversed this association, i.e., by pressing "f" in response to movements with lower peak velocity and "n" in response to movements with higher peak velocity.

Two dependent variables were computed from participants' number of Iconic and Non-Iconic mappings (aggregated across all six blocks): Calculating the *absolute* difference between the total number of Iconic mappings and the total number of Non-Iconic mappings, separately for each goal type condition and each degree of exaggeration, gave us a *Consistency score* for each participant ranging from 0 to 12. A Consistency score of 0 meant that participants mapped velocities randomly to targets and a score of 12 meant that participants mapped with absolute consistency. Calculating the *signed* difference between Iconic and Non-Iconic mappings gave us the *Mapping score*, which could range from +12 (fully iconic mappings) to -12 (fully non-iconic mappings). A Mapping score of 0 meant that participants lacked a preference for Iconic or Non-iconic mappings.

Participants who pressed the same key (either "n" or "f") at least ten times in a row were excluded from further analysis. Based on this criterion, one participant was excluded in Experiment 1.

**Mapping consistency.** To test whether participants interpreted the observed velocity differences in a consistent manner, we compared the distribution of Consistency scores to 0 (i.e., inconsistent mapping) using separate Bonferroni-corrected one-sample $t$-tests. Consistency scores are displayed in Fig 3, where each dot represents an individual participant grouped according to degrees of exaggeration. The scores differed significantly from 0 across all degrees of exaggeration and goal type (all $t(23) > 6.4$, $p < .001$, $d > 1.3$, one-tailed). This result shows that participants were able to distinguish the different animated movements in terms of velocity and, thereby, to consistently map them to either the near or the far target location,

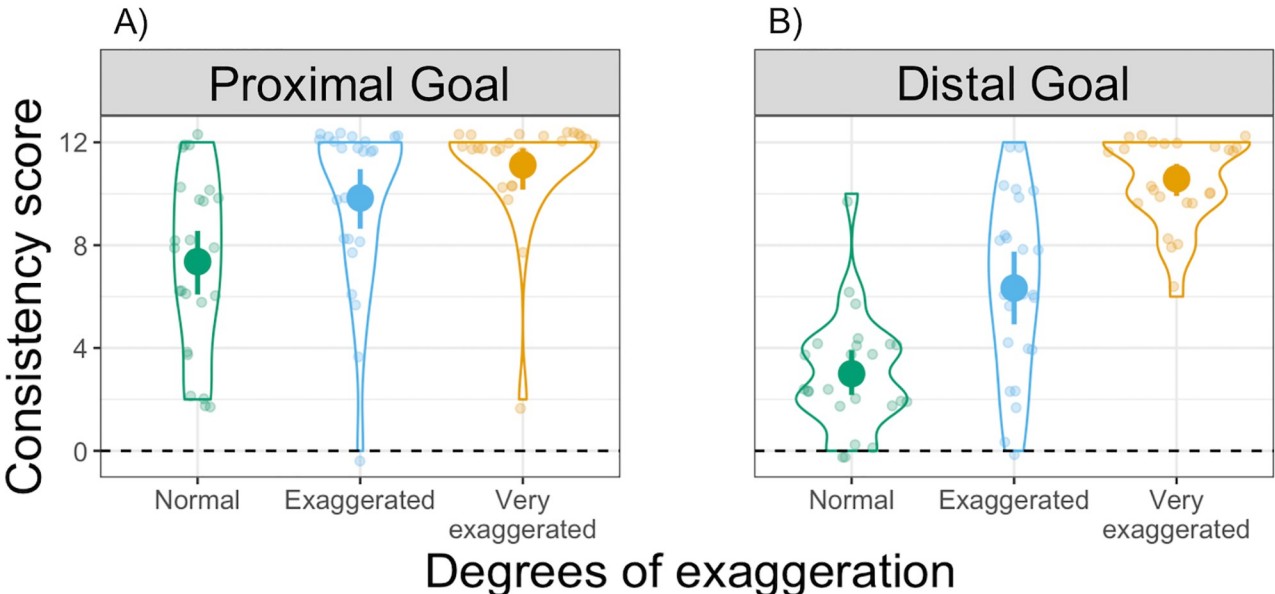

**Fig 3. Consistency scores in Experiment 1.** Distribution of Consistency scores in the (A) Proximal goal and (B) Distal goal conditions. Each dot represents an individual participant, with one Consistency score for each degree of exaggeration: Normal in green, Exaggerated in light blue and Very exaggerated in yellow. Violin plots represent the overall distribution of Consistency scores for each degree of exaggeration. The dashed horizontal line indicates a hypothetical value for random mapping (i.e., no consistency).

regardless of how exaggerated the velocity profile was and regardless of whether the movement achieved a proximal goal or a distal goal.

To address the role of exaggeration and goal type, we conducted a 2x3 ANOVA with Consistency scores as dependent variable, goal type (Proximal and Distal goal) as between-subject variable and degrees of exaggeration (Normal, Exaggerated, Very exaggerated) as within-subject variable. We found a significant main effect of goal type ($F(1,47) = 23.6$, $p < .001$, $\eta_p^2 = .21$) and a significant main effect of degrees of exaggeration ($F(2,94) = 71.1$, $p < .001$, $\eta_p^2 = .42$). There was also a significant interaction between these factors ($F(2,94) = 8.9$, $p < .001$, $\eta_p^2 = .08$). Pairwise comparisons using Bonferroni-corrected $t$-tests within the Proximal goal condition showed significant differences between non-exaggerated (i.e., Normal) and both exaggerated movements (Exaggerated: $t(94) = -3.7$, $p = .001$, $d = 0.76$; Very exaggerated: $t(94) = -5.6$, $p < .001$, $d = 1.3$). In the Distal goal condition, all pairwise comparisons between degrees of exaggeration yielded significant differences (all $t(94) < -4.9$, $p < .001$, $d > 1.1$). These results show that the larger the differences in movement velocities, the more consistently participants mapped them to the respective target location.

**Mapping score.** To investigate whether participants were more likely to map movements to targets in line with the motor-iconic prediction, we computed separate Bonferroni-corrected one-sample $t$-tests comparing the Mapping scores of each condition to 0 (i.e., random mapping direction). Mapping scores are displayed in Fig 4, where each dot represents an individual participant's Mapping score for each degree of exaggeration. We found that in the Proximal goal condition participants' responses were significantly different from chance (all $t(24) > 9.7$, $p < .001$, $d > 1.95$), showing a clear preference for Iconic mappings; see Fig 4A. In the Distal goal condition, however, participants' responses did not differ significantly from chance (all $t(23) > 0.21$, $p > .08$, $d > 0.04$). Thus, in the Distal goal condition, individual participants overall used a consistent mapping (resulting in a Consistency score that significantly differed from chance, as reported above), yet, across participants, there was no complete conformity as

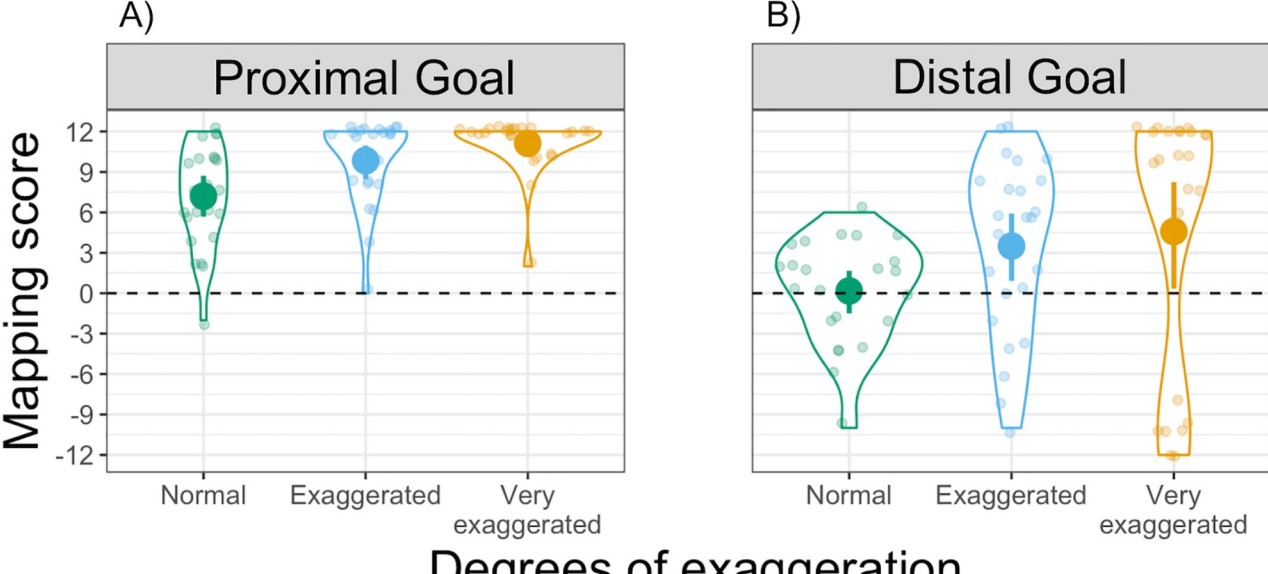

**Fig 4. Mapping scores in Experiment 1.** Distribution of Mapping scores in the (A) Proximal and (B) Distal goal conditions, for the three degrees of exaggeration. Each dot represents an individual participant, with one Mapping score for each degree of exaggeration. Violin plots represent the overall distribution of Mapping scores for each degree of exaggeration. The dashed horizontal line indicates random mapping direction.

to the direction of that mapping (i.e., whether to map faster movements to the far target location and slower movements to the near target location or vice versa).

We conducted a 2x3 ANOVA with Mapping scores as dependent variable, goal type as between-subject variable and degrees of exaggeration as within-subject variable. The ANOVA yielded a significant main effect of goal type ($F(1,47) = 26.7$, $p < .001$, $\eta_p^2 = .28$), as well as a significant main effect of degrees of exaggeration ($F(2,94) = 15.5$, $p < .001$, $\eta_p^2 = .09$). The interaction between these two factors, however, was not significant ($F(2,94) = 0.1$, $p = .86$, $\eta_p^2 < .001$). Bonferroni-corrected post-hoc $t$-tests were used to analyze the main effect of exaggeration. In the Proximal goal condition, only the comparison between Normal and Very exaggerated yielded a significant result ($t(94) = -3.6$, $p = .001$, $d = 1.3$), whereas in the Distal goal condition the comparison between Normal and both exaggerated movements yielded significant results (Exaggerated: $t(94) = -3.0$, $p = .009$, $d = 0.6$; Very exaggerated: $t(94) = -4.0$, $p < .001$, $d = 0.6$).

## Discussion

We hypothesized that participants would detect differences in the velocity of the observed movements and, based on these differences, consistently map the movements to one of the two target locations, particularly when the velocity differences were exaggerated. Our results support this hypothesis: Consistency scores significantly differed from chance and they were higher for more exaggerated movements.

In line with previous findings showing that movement velocity and distance are systematically related in natural aiming movements [39], we had further hypothesized that participants would map slower movements to near target locations and faster movements to far target locations. Since these mappings preserve the relationship between movements and their most likely goals, they can be understood as "motor-iconic". Our results in the Proximal goal condition provide evidence in support of this hypothesis, as shown by participants' preference for Iconic mappings in this condition. The extent to which participants produced these mappings was also strongly influenced by the degree of exaggeration, as more exaggerated movements led the majority of participants to produce more Iconic mappings.

Our interpretation of this finding in terms of motor-iconicity implies that participants directly 'perceive' the observed movements in terms of their proximal goals and that it is not a learned mapping. However, we need to acknowledge that the structure of our familiarization could, in principle, have contributed to the strong preference towards motor-iconic mappings in the Proximal condition. Participants were familiarized with two fully visible movements, each directed at one of the two target locations, and these two movements contained actual differences in their peak velocities. That means that, although both movements were Normal (i.e., non-exaggerated), the movement with the slightly higher peak velocity was directed at the far target and the movement with the slightly slower peak velocity was directed at the near target (see Fig 2A). This was a natural consequence of our decision to present real movements during familiarization (and not, e.g., movements artificially made equal). However, given that the familiarization only contained two trials that introduced participants to the details of the overall procedure, we consider it unlikely that this created a strong bias in participants' responses.

Altogether, the findings of the Proximal goal condition are consistent with previous research showing that observers can derive useful anticipatory information from partially occluded actions and use this information to derive their partner's proximal goals in a joint setting [10].

Our results in the Distal goal condition indicate that movement modulations can also be used to extract information about an upcoming, more distal goal. Our findings in this condition show that, even when movements are not exaggerated, participants still produce a higher than chance rate of consistent movement-to-location mappings (see Consistency score, Fig 3B).

However, when looking at the direction of these mappings (see Mapping score, Fig 4B), we found that there was no clear preference towards either of the two potential mapping directions in this condition: independent of exaggeration, about half of the participants chose to map faster movements onto near target locations and slower movements onto far target locations.

More generally, our pattern of results also seems to suggest that, depending on whether the observed movement reaches a proximal or a distal goal, participants might be resorting to different strategies to interpret the movements they see. When the movement attains only a proximal goal, as in the Proximal goal condition, participants need to observe the moving box and simulate its more likely goal given its velocity before occlusion [46, 47]. In such circumstances, participants can readily identify the motor-iconic relationship connecting the movements and the target locations, i.e., faster/slower movements leading to farther/nearer targets. When the movement achieves a distal goal, as in the Distal goal condition, participants cannot simply extrapolate the observed sliding movement towards the targets, as the box stops moving when it reaches the gray target in the middle of the screen. This is then followed by a sudden disappearance of the box from the display. As a consequence of this, the underlying motor-iconic relationship between the movement and its distal goal may not have been recognized in the Distal goal condition, as shown by the fact that participants, collectively, did not display a clear preference for Iconic over Non-Iconic mappings. To what extent is this lack of a general preference due to the spatiotemporal separation between a movement and a distal goal, or specifically to the fact that such separation was introduced in the present study by means of a sudden teleportation, is at present not clear. However, when asked to describe the strategy they used to solve the task, none of our participants in the Distal goal condition mentioned anything about the teleportation, while most of them made some reference to the difference in the velocity of the proximal movement, either in ways that are consistent with a motor-iconic interpretation (e.g., *My strategy depended on the speed of the box. Slow for near, fast for far*), or with its reversal (e.g., *If it looked like the box moved fast I selected NEAR*). This seems to suggest that participants were indeed able to interpret the relationship between the movements and their distal goals, despite the fact that the achievement of this latter goal was made possible by means of a teleportation of the box.

An alternative to the above interpretations could be that participants simply had problems to perceptually distinguish the velocity differences in the Distal goal condition. Although the high Consistency scores suggest that participants were able to discriminate between fast and slow movements, we conducted Experiment 2 to safely exclude the possibility that the pattern of results found in the Distal goal condition is due to difficulties in perceptually discriminating the different movement velocities.

## Experiment 2: Perceiving velocity modulations

To determine whether the pattern of results found in the Distal goal condition really reflects an uncertainty about how to map the perceived velocity differences onto the occluded target locations, or whether it is simply due to difficulties in perceptually discriminating movements of different velocity, we conducted Experiment 2. Participants were shown the same animations as in Experiment 1, but were now asked to determine whether the movements were fast or slow. We predicted that participants would be able to discriminate the movements and that discrimination performance would be better for more exaggerated movements.

### Methods

**Participants.** We recruited 49 participants (21 women; Age: $M$ = 28.7 years; $SD$ = 8.2 years) through Testable. The conditions of recruitment were identical to Experiment 1.

**Stimuli, design, & procedure.** Participants were presented with exactly the same animated movements as in Experiment 1, but their task was now to identify whether the movements were fast (by pressing the "f" key) or slow (by pressing the "s" key). As in Experiment 1, roughly half of the participants took part in the Proximal goal condition, the other half in the Distal goal condition. The only difference to Experiment 1 concerned the familiarization, where participants saw the occluded scene right away, and consequently never saw the two target locations on the right side of the screen. This choice was made to have participants focus on the velocity differences without making implicit associations about movement distance. As in Experiment 1, participants did not receive any kind of accuracy feedback.

## Results

**Data preparation.** From participants' individual responses, we counted the total number of correct and incorrect responses for each movement, depending on the degree of exaggeration and the type of goal. We then subtracted these two values to obtain a *Discriminability score* that ranged from +12 (fully correct discrimination) to -12 (fully incorrect discrimination), which we could use to directly compare the results of Experiment 2 with the Mapping scores of Experiment 1.

**Movement discrimination.** The Discriminability score enabled us to measure the extent to which participants were able to correctly identify and categorize the movements as slow or fast on the basis of their differences in velocity. As expected, participants were able to correctly discriminate the movements, as shown by the significant difference from chance (i.e., higher than 0, see Fig 5) when movements were Normal, Exaggerated or Very exaggerated in both goal type conditions (all $t(23) > 5.1$, $p < .001$, $d > 1.0$, one-tailed).

**Comparison across experiments.** To test whether the task (mapping different velocities to different target distances vs. discriminating different velocities) had an impact on

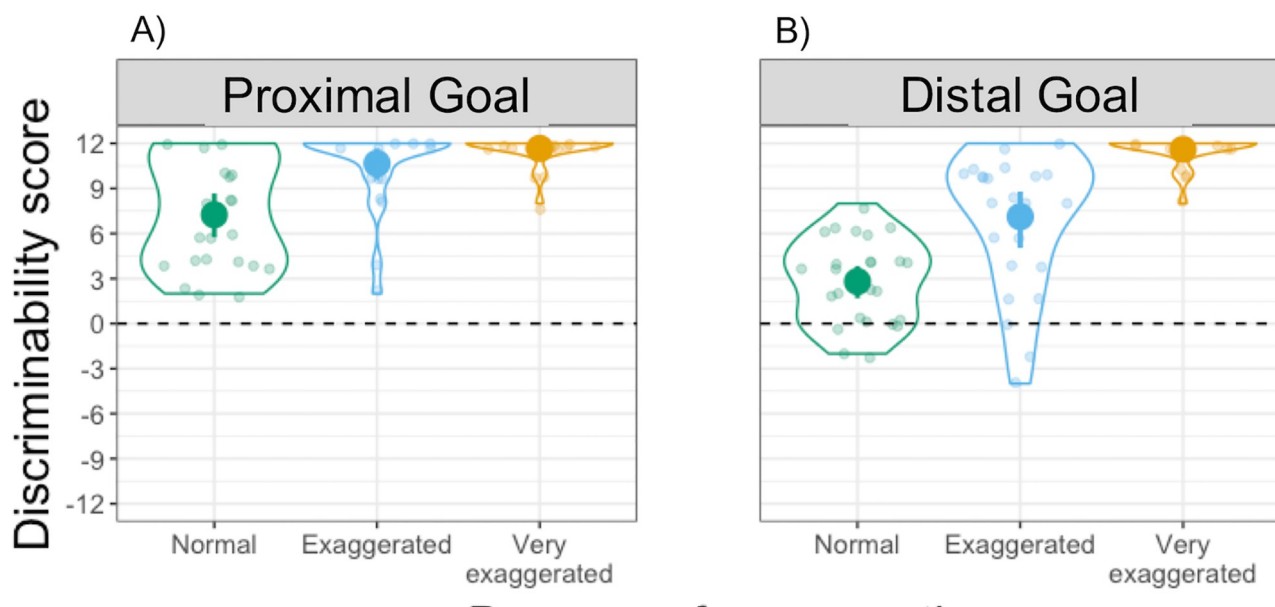

**Fig 5. Discriminability scores in Experiment 2.** Distribution of Discriminability scores in the (A) Proximal goal and the (B) Distal goal conditions, for the three degrees of exaggeration. The dashed line indicates chance discriminability (i.e., that movement velocities are not discriminable).

participants' behavior, we conducted a 2x2x3 ANOVA comparing the Mapping scores of Experiment 1 and the Discriminability scores of Experiment 2 in both goal type conditions and across all three levels of exaggeration. The ANOVA yielded significant main effects of Experiment ($F(1,94) = 12.5$, $p < .001$, $\eta_p^2 = .08$), goal type ($F(1,94) = 45.2$, $p < .001$, $\eta_p^2 = .2$), and degrees of exaggeration ($F(2,188) = 65.3$, $p < .001$, $\eta_p^2 = .2$). We also found significant interactions between Experiment and goal type ($F(1,94) = 8.1$, $p = .005$, $\eta_p^2 = .05$), Experiment and degrees of exaggeration ($F(2,188) = 3.3$, $p = .04$, $\eta_p^2 = .01$) and goal type and degrees of exaggeration ($F(2,188) = 3.4$, $p = .04$, $\eta_p^2 = .01$). To further explore the main effects, we conducted Bonferroni-corrected post-hoc $t$-tests comparing participants' Mapping and Discriminability scores across the two Experiments for each degree of exaggeration. The analyses revealed significant differences between participants' responses across the two Experiments in the Distal goal condition, regardless of the degree of exaggeration (all $t(209) > -5.6$, $p > .04$, $d > 0.6$). In the Proximal goal condition, however, none of the pairwise comparisons across Experiments yielded a significant result (all $t(209) > -0.6$, $p > .5$, $d < 0.3$).

## Discussion

The results of Experiment 2 indicate that participants can discriminate between the different velocities at a higher than chance level in both goal conditions. Along with the high Consistency scores of Experiment 1, these findings suggest that participants in the Distal goal condition are able to correctly distinguish the different movement velocities, and that their relatively lower Consistency scores and lack of preference for a unique mapping direction in Experiment 1 are not due to difficulties in perceptual discrimination. Instead, it seems that modulations in velocity, even when they are correctly categorized as fast or slow, are not uniformly associated to a unique distal goal in this condition.

Taken together, the results of the two experiments raise the possibility that some participants in the Distal goal condition in Experiment 1 might not have been aware of the underlying motor-iconic relation between the proximal movement and the distal goal. Specifically, participants who opted for Non-iconic mappings might not have been influenced by the intrinsic link between movement velocity and target distance [39] that characterizes human natural performance. In order to address this hypothesis directly, we conducted Experiment 3.

## Experiment 3: Interpreting goal type

Participants in Experiment 3 were presented with either Proximal goal trials followed by Distal goal trials, or vice-versa. We reasoned that being presented with trials where the movement attains its proximal goal first might subsequently help participants to recognize the underlying motor-iconic relation between velocity and distal goal. If that was the case, then we should expect participants who are initially presented with Proximal goal trials, followed by Distal goal trials, to apply the underlying connection recognized during the Proximal goal trials also to the Distal goal trials. We also presented another group of participants with Distal goal trials followed by Proximal goal trials, and reasoned that those participants who disregard the motor-iconic relation during the Distal goal trials might also disregard it later on, during the Proximal goal trials. This would suggest that establishing a particular interpretation of the movements early on might have the effect of overruling the effects of the motor-iconic relation.

### Methods

**Participants.**   We recruited 100 participants (35 women; Age: $M = 29.6$ years; $SD = 9.7$ years) through Testable. The conditions of recruitment were identical to Experiments 1 and 2.

**Stimuli, design, & procedure.** Participants were presented with the same animated movements as in Experiment 1 and their task was, as in Experiment 1, to choose the likely target location for each movement. This time, however, participants were randomly assigned to one of four conditions. The Proximal goal and Distal goal conditions were identical to the Proximal goal and Distal goal conditions of Experiment 1. The only difference pertained to the instructions as explained below. The other two conditions were a mix: Participants saw either three blocks of Proximal goal followed by three blocks of Distal goal trials (i.e., PG-to-DG condition), or three blocks of Distal goal followed by three blocks of Proximal goal (i.e., DG-to-PG condition).

During familiarization, participants were informed that the experiment consisted of two parts, and that their task would be to guess to which target location the box was being delivered. Halfway through the experiment, participants were introduced and familiarized with the second part. For participants in the Proximal goal and Distal goal conditions, this second familiarization was identical to the one they saw at the beginning. For those in the PG-to-DG and the DG-to-PG conditions, the new familiarization introduced the new layout, corresponding to what participants would see during the second part of the study (i.e., Distal goal in the PG-to-DG condition and Proximal goal in the DG-to-PG condition). In all conditions, participants were told that the movements they would see during the second part of the experiment had been recorded from another participant than those during the first part.

## Results

**Data preparation.** As in Experiment 1, we computed Consistency and Mapping scores for each participant across all three degrees of exaggeration. In order to render the data comparable across stable and mixed conditions, we computed separate scores for the first and second half of the experiment. As a consequence, the Consistency scores for each participant now ranged from 0 to 6 (Fig 6), and the Mapping scores ranged from -6 to +6 (Fig 7). Six participants who pressed the same key ten times in a row were excluded from the analyses.

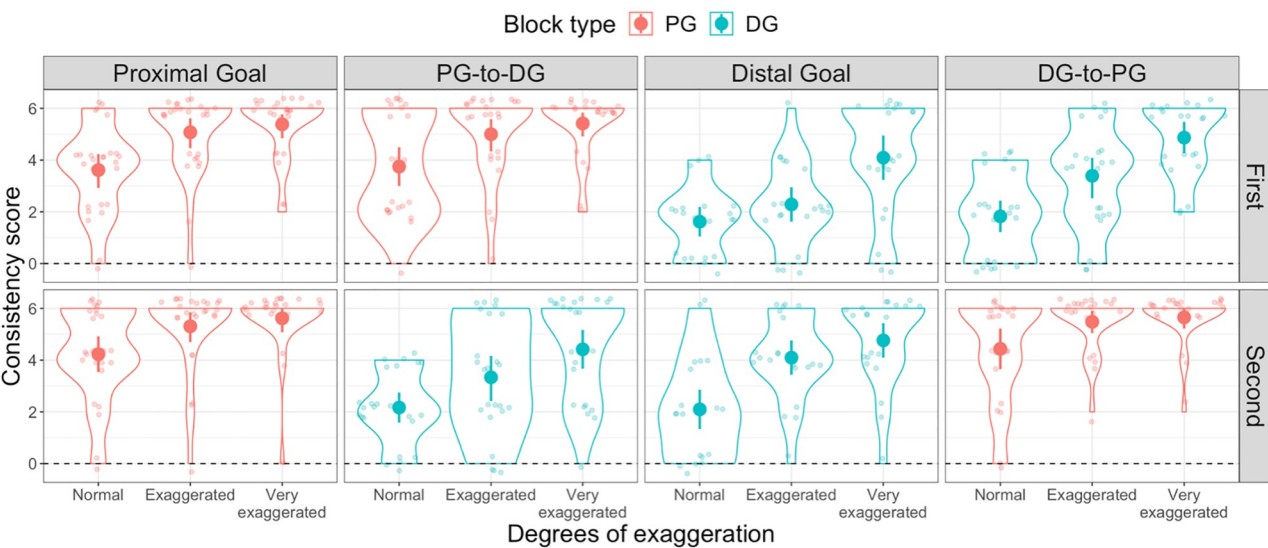

**Fig 6. Consistency scores in Experiment 3.** Distribution of Consistency scores in the (from left to right panel) Proximal goal, PG-to-DG, Distal goal and DG-to-PG condition. The upper panels display the distribution in the first half of the Experiment, while the lower panels display the distribution in the second half of the Experiment, for all three degrees of exaggeration. Violin plots in red represent Proximal goal trials, while violin plots in light blue represent Distal goal trials. The dashed line indicates random mapping (i.e., no consistency).

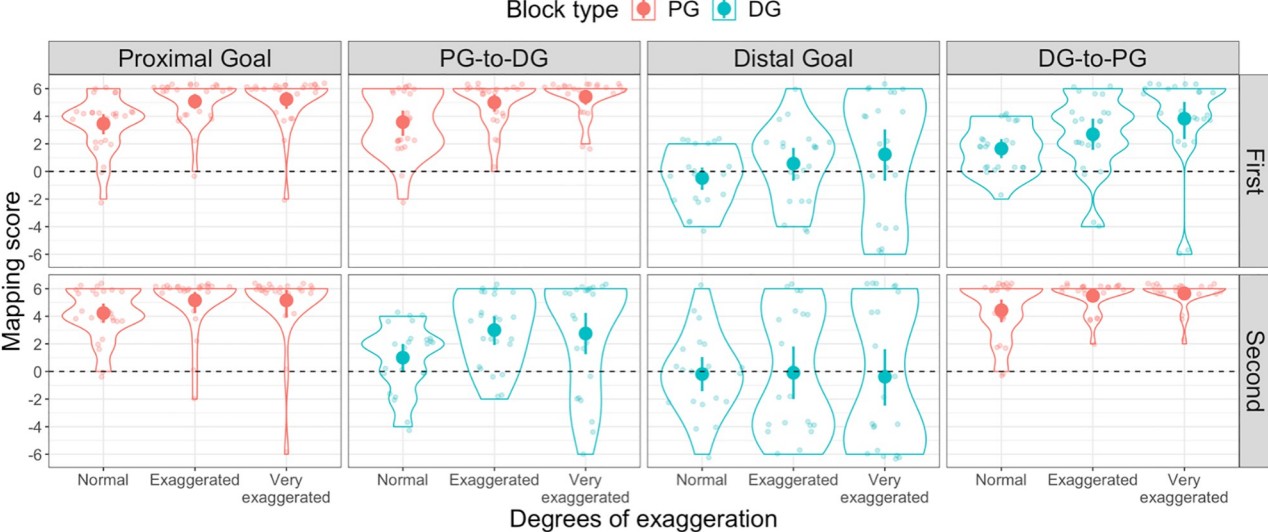

**Fig 7. Mapping scores in Experiment 3.** Distribution of Mapping scores in the (from left to right panel) Proximal goal, PG-to-DG, Distal goal and DG-to-PG condition. The upper panels display the distribution of Mapping scores of the first half of the Experiment, while the lower panels display the distribution of Mapping scores of the second half of the Experiment, for all three degrees of exaggeration. Violin plots in red represent trials in which participants were presented with Proximal goal trials, while violin plots in light blue represent trials in which participants were presented with Distal goal trials. The dashed line indicates random mapping direction.

## Mapping consistency and direction

We found that participants were able to produce consistent mappings across all degrees of exaggeration and goal type, as shown by the significant differences from a non-consistent baseline (i.e. 0) (all $t(20) > 4.9$, $p < .001$, $d > 1$, one-tailed) in both the first and second half of the experiment (see Fig 6). Regarding the direction of the mappings, we found that participants in the stable Distal goal condition were, collectively, equally likely to produce either Iconic or Non-Iconic mappings, as indicated by the non-significant difference from chance in that condition across all three degrees of exaggeration (all $t(20) > -1.0$, $p = 1$, $d > 0.02$, two-tailed), thus replicating our results from Experiment 1. Surprisingly, we did not find this pattern of results in the first half of the DG-to-PG condition, where participants were more likely to produce Iconic mappings (all $t(22) > 4.4$, $p < .004$, $d > 0.93$, two-tailed). This preference for Iconic mappings was also present in most other conditions (all $t(23) > 7.7$, $p > .001$, $d > 1.5$). The only two exceptions to this pattern of results were the previously mentioned Distal goal condition and the second half of the PG-to-DG condition in which our group of participants failed to display a clear preference towards any of the two mapping directions when the movements were either Normal ($t(23) = 2$, $p = 1$, $d = .4$) or Very exaggerated ($t(23) = 3.3$, $p = .06$, $d = .07$) (see Fig 7).

**Effect of previously viewed trials.** To assess the effects of having been presented with a Distal or Proximal goal on participants' Mapping scores during the second half of the experiment we computed two ANOVAs. The first ANOVA compared the Mapping scores of the second half of the stable Proximal goal condition to those of the second half of the DG-to-PG condition. The ANOVA only yielded a main effect of degrees of exaggeration ($F(2,94) = 10.6$, $p < .001$, $\eta_p^2 = .07$), but no main effect of condition ($F(1,47) = .59$, $p = .45$, $\eta_p^2 = .008$). The second ANOVA compared the Mapping scores of the second half of the stable Distal goal condition to those of the second half of the PG-to-DG condition. The ANOVA yielded only a main effect of condition ($F(1,43) = 7.4$, $p = .009$, $\eta_p^2 = .1$). Bonferroni-corrected post-hoc $t$-tests

revealed that Mapping scores were significantly higher in the second half of the PG-to-DG condition, compared to the second half of the Distal goal condition, but only when movements were exaggerated (Exaggerated: $t(85) = 2.8$, $p = .006$, $d = .8$; Very exaggerated: $t(85) = 2.8$, $p = .005$, $d = .6$).

**Distal goal condition across experiments.**   In order to gain further insight into why participants' Mapping scores differ between the first half of the DG-to-PG condition and the first half of the Distal goal condition we compared these two conditions to the first half of the Distal goal condition of Experiment 1 using Bonferroni-corrected $t$-tests. The results revealed that, when comparing the Mapping scores of the Distal goal condition of Experiment 1 to those of the Distal goal condition of Experiment 3, none of the tests yielded significant differences across any of the three degrees of exaggeration (all $t(146) > 0.4$, $p > .7$, $d < 0.37$). No significant differences were found when comparing the first half of the No overlap condition of Experiment 1 to the first half of the DG-to-PG condition of Experiment 3 (all $t(146) > -1.8$, $p > .2$, $d < 0.77$) either.

## Discussion

Experiment 3 showed that, as we predicted, being first presented with a situation in which the movement achieves a proximal goal (i.e., Proximal goal trials), helps participants recognize the underlying connection between movement velocity and distance, and that, once they recognize it, they can apply it to a situation in which the movement achieves a distal goal (i.e., Distal goal trials). This seems to suggest that the pattern of results found in Experiment 1, where some participants in the Distal goal condition preferred to use Non-Iconic mappings, is at least partially due to the fact that participants in that condition were not taking into consideration the underlying, motor-iconic relation that connects velocity to spatial locations. In Experiment 3 we addressed this hypothesis by assigning participants to a condition in which the connection between movement and target location was made explicit during the first half of the experiment, the PG-to-DG condition, and predicted that this would have an effect on participants' responses, specifically on their Mapping scores, during the second half of the experiment. The results in this condition support this hypothesis, as participants were more likely to produce Iconic mappings in the latter part of the experiment, right after being presented with trials in which the connection between the movements and their target locations was made explicit.

Our results also suggest that, although the explicit connection between movement and target location that was established thanks to trials in which the movements achieved a proximal goal has an effect on participant's responses, this precedence effect does not hold when we reversed the order of the trials, i.e., in the DG-to-PG condition. Specifically, participants in this condition, who were first presented with trials in which movements achieved a distal goal, still produced Iconic mappings consistently during the second half of the experiment (i.e., during the Proximal goal trials) in a way that was almost indistinguishable from that found in the Proximal goal condition.

In sum, our results in Experiment 3 point to the possibility that observing a movement that achieves a distal goal might be introducing, across participants, a relatively weak preference for any of the two possible mappings (i.e., Iconic or Non-Iconic). This can be seen in the lack of a general mapping preference in the Distal goal condition of Experiments 1 and 3. Such lack of a general preference for a specific mapping might also explain the differences we found between the first half of the DG-to-PG condition and the Distal goal condition. Although identical during the first half of the experiment, these two conditions surprisingly led to differences in participant's Mapping scores. As our analyses suggest, this pattern of results might be simply due

to the fact that in these two conditions our group of participants was either more likely than average to lack a preference towards any of the two mapping directions (i.e., the Distal goal condition of Experiment 3) or had a stronger preference than average to produce Iconic mappings (i.e., the first half of the DG-to-PG condition), or a combination of the two. In sum, this supports the idea that observing a movement achieving a distal goal weakens the preference for a particular mapping direction among some participants. This, in turn, might explain the differences found across the Distal goal and DG-to-PG conditions.

## General discussion

The aim of the present study was to investigate whether observers can interpret proximal communicative actions in terms of their distal goals. We hypothesized that participants would be able to detect communicative modulations of movement velocity and consistently interpret them in terms of the actions' distal goals (i.e., target locations), even though those distal goals could never be directly observed.

Our findings support this hypothesis, providing first evidence that observers can derive information about both proximal and distal goals from simple, one-dimensional movements. Specifically, participants in Experiment 1 were able to infer the likely proximal and distal goals of an action by relying on differences in movement velocity. In this respect, our findings are consistent with previous computational work suggesting that modulations of early kinematic features can be used strategically to communicate and disambiguate between an actor's distal goals [35]. Additionally, our results show that participants benefited from exaggerated velocity differences, allowing them to produce more consistent mappings. From this perspective, the present study provides further support to the previously established finding in SMC that observers can predict their partner's upcoming actions by relying on subtle kinematic modulations in their goal-directed movements [9–13], while at the same time making a novel contribution by extending these findings to a setting where observers need to infer an actor's distal goals.

Moreover, our results provide a first empirical demonstration that the way observers interpret communicative modulations is in part affected by whether the observed movement achieves a proximal goal only, or achieves a distal goal by means of attaining a more proximal goal first. When the movement achieves a proximal goal, observers display a clear preference towards interpreting the communicative modulations of that same movement in a motor-iconic manner, that is, in a manner that preserves the underlying relation between the movement and its likely goal. Although not completely absent, this preference for a motor-iconic relation is reduced when the movement achieves a distal goal. This might be partially due to the fact that participants fail to see the connection between a proximal movement and its more distal goal, especially when the transition between movement and goal is established indirectly by means of an unexpected event (a teleportation, as is the case in the present study). Understanding the factors that increase or decrease the preference for motor-iconic mappings when movements provide information about distal goals remains an important question for future research.

One hypothesis for why reference to distal goals reduces the preference for motor iconic-mappings is that seeing movements that achieve distal goals might induce a different, possibly more arbitrary or symbolic interpretation of the communicative modulations than seeing movements directed to proximal goals. Within the context of SMC, arbitrary mappings have been reported previously in studies in which participants were asked to coordinate their actions in tasks in which "Leader" participants with task-relevant information about specific target locations could opt to communicate this information to their naïve partners by means

of either exaggerating the kinematics of their goal-directed movements (e.g., movement duration), thus resulting in what we here refer to as motor-iconic mappings, or alternatively by creating stable associations between non-dynamic features of these same movements (e.g., the dwell time on a target location) and the different target locations [13]. These latter associations were described as "symbolic", as there seems to be no underlying motor relationship between the dwell time on a target location and the relative distance of that same target location. Interestingly, by switching from a more iconic to a more symbolic form of communication, actors were also able to create a temporal and functional separation between the instrumental (i.e. moving towards the target) and communicative (i.e. informing their partner) aspects of their movements, which in turn provided a more efficient way of informing their partners about their goals. Our Distal goal condition can be understood as an extension of these previous results, in that the achievement of a distal goal by means of attaining a more proximal one first may have also been interpreted by participants as a separation between the more instrumental aspect of the movement (i.e. delivering the box towards one of the target locations) and a more communicative one (i.e. informing an observer about a correct target location). As a consequence, the finding that participants displayed different mapping preferences in the Distal goal condition may be partly due to the fact that they took the first sliding movement to be a purely communicative movement, akin to a communicative gesture like pointing, with no explicit connection to an instrumental goal, let alone a distal one.

Relatedly, the strong preference for motor-iconic mappings in the Proximal goal condition raise the question of whether participants might be deriving the relationship between movement velocity and distance in a way that resembles an indexical relation, rather than a motor-iconic one. Indices are often described in semiotic theory as signs that carry information by virtue of them having an intimate relation with the objects they indicate, either in terms of spatio-temporal contiguity (e.g., a pointing gesture) or in terms of causality (e.g., smoke as a sign of fire) [48]. As we noted previously, there is a lawful relationship between the velocity and the distance of aiming movements during natural (i.e., non-exaggerated) performance, such that proximal movements directed at far target locations are performed with higher peak velocities [39]. This same intimate relation between velocities and distance was found in the way observers infer proximal goals from communicative movement modulations, as shown in the Proximal goal condition. Thus, it is a possibility that participants in that condition simply extracted some form of causal regularity or spatiotemporal contiguity that connects movement velocity and distance, thereby making velocity an indexical sign for proximal goals. From this perspective, introducing an explicit spatial and temporal separation between a proximal movement and its goal, as we did in the Distal goal condition, may have concurrently led to a change in the way participants derived information from the movements they observed, from a purely indexical interpretation to a more iconic one. This shift in the relation between signs and referents is similar to the one described by Keller, who illustrates this effect with the example of a real and a simulated yawn [49]. Whereas a real yawn is simply taken as a sign caused by an underlying physiological state (e.g., tiredness) and is therefore an index, a simulated yawn is not, since it lacks, by definition, such causal antecedent. Instead, simulated yawns, because they are exaggerated simulations of the real yawn but still resemble it in some relevant way, are better described as icons. Similarly, movements that are directed at achieving distal goals but whose kinematic features resemble in some informative way those directed at proximal ones might also be taken as the iconic extension of the more basic, indexical causal relationship that connects proximal movements to their proximal goals. Under this view, an iconic interpretation is at least partly grounded on more basic indexical one, and therefore might explain the strong precedence effect that observing movements achieving their proximal goal has on subsequent interpretation of movements achieving a distal goal, as we found in Experiment 3.

The above discussion also points to the possibility that participants in the Distal goal condition may have thought of the sliding movement towards the middle of the screen as the first movement component of an implied two-step action sequence. Although participants in the Distal goal condition were only presented with a single sliding box movement, followed by the sudden disappearance of the box, we cannot discard the possibility that they might have tried to substitute the perceptual gap introduced by this sudden "teleportation" by simulating a second sliding movement, similar to the one they had just seen [46, 47]. Whether participants did indeed generate such simulations, and whether these simulations affected the way they mapped the movements onto the target locations, are both open questions for further research.

Given our initial interest in addressing questions related to the understanding of communicative modulations, our study focused exclusively on the receiver end of the interaction, that is, on whether and how observers interpret proximal communicative actions in terms of their proximal and distal goals. Moreover, as the movements participants saw in the present study were created for the purpose of the experiment, the question remains whether participants who had the task to actively inform someone else using our setup would display a spontaneous preference towards creating mappings that preserve the motor-iconic relation between proximal movements and distal goals (i.e., Iconic mappings), or would rather opt to reverse this mapping (i.e., Non-Iconic mappings). Alternatively, communicators might opt to convey such information through other relevant movement parameters, either continuous ones as when people directly modulate the total duration of their movements while trying to keep the velocity constant, or discrete ones such as movement pauses or sudden changes in movement direction. These different strategies are not a mere theoretical possibility, since they correspond to what some participants in our studies reported at the end of the experiment when they were asked which communicative strategy they would choose. While the majority described strategies that were consistent with a motor-iconic one (e.g., "*I'd move it very slowly for the near location and very quickly for the far location*") or with its reversal (e.g., *I would move the box slower to reach the further location*), a few did mention other movement parameters (e.g., jerk: "*I would move it jerkily–one jerk for near and two for far*"; deceleration: "*Ending the placement of [the box] faster for far [. . .] slower for near.*").

A final consideration concerns the role of objects used in order to convey information about action goals. Our study presented participants with animations of a moving box with a cursor attached to it and a context in which this box was smoothly "delivered" along a black line towards one of two target locations. By doing this, we tried as much as possible to highlight the relevant differences in the velocity of the box, while also minimizing other factors, such as the weight or fragility of the box, that could have had an influence on participant's interpretations. For example, one would normally expect a heavy or fragile object to be moved slowly in order to maintain control during its transport and placement. If applied to our task, this would mean that an actor would be forced to adopt a slower pace when sliding the box, thus having a direct effect on the options of communicative strategies at her disposal. Again, future studies could explore the role that objects, with their specific physical properties and affordances, might play in the way communicators flexibly adapt their modulations to provide information to observers [37].

The present study offers valuable perspectives for future research on joint action and communication. As argued at the outset, it is likely that people engaged in a joint action will try to predict their partner's distal goals by relying on a wide variety of kinematic cues. Being able to make such long-term predictions can be particularly useful in situations where co-actors produce complex action sequences that require the coordination of actions at different temporal levels [50], such as dancing or playing football. In such scenarios, providing relevant

information about the upcoming distal goals early on in the action sequence would be a useful and effective manner to facilitate coordination.

## Supporting information

**S1 Appendix. Movement recording and exaggeration.** A detailed description of the procedure used to record, exaggerate and rescale the movements.
(DOCX)

**S2 Appendix. Control condition.** Description and analyses of a control condition in Experiment 1.
(DOCX)

## Acknowledgments

We thank Fanni Takátsy for her help with data collection.

## Author Contributions

**Conceptualization:** Martin Dockendorff, Laura Schmitz, Cordula Vesper, Günther Knoblich.

**Data curation:** Martin Dockendorff.

**Formal analysis:** Martin Dockendorff.

**Funding acquisition:** Günther Knoblich.

**Investigation:** Martin Dockendorff, Laura Schmitz, Cordula Vesper, Günther Knoblich.

**Methodology:** Martin Dockendorff, Laura Schmitz, Cordula Vesper, Günther Knoblich.

**Project administration:** Günther Knoblich.

**Resources:** Günther Knoblich.

**Software:** Martin Dockendorff.

**Supervision:** Cordula Vesper, Günther Knoblich.

**Validation:** Martin Dockendorff.

**Visualization:** Martin Dockendorff.

**Writing – original draft:** Martin Dockendorff.

**Writing – review & editing:** Laura Schmitz, Cordula Vesper, Günther Knoblich.

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
