## [Decision Letter · Decision Letter 0]

14 Sep 2022

PONE-D-22-13033

Understanding others’ distal goals from proximal communicative actions

PLOS ONE

Dear Dr. Dockendorff,

I have received two expert reviews on your paper "Understanding others’ distal goals from proximal communicative actions", which you submitted to PLOS ONE.

As you will see from the reviews attached to the end of this letter, both reviewers found merit in your work, but they also identified a number of issues that need more work. Their comments are nicely detailed and I agree with their assessment. An important issue pointed out by reviewer 1 is whether inferences are truly communicative. Both reviewers also point the need for more details regarding the methods in order to better grasp how participants interpreted actions. Therefore, I invite you to submit a revision together with a cover letter explaining how you have responded to the reviewers’ comments. Should you resubmit a revision, I will send the new version to some or all of the same reviewers.

Sincerely

We look forward to receiving your revised manuscript.

Kind regards,

Cédric A. Bouquet

Academic Editor

PLOS ONE

“This research was supported by the European Research Council under the European Union's Seventh Framework Program (FP7/2007-2013) / ERC grant agreement n° [609819], 961 SOMICS”

“This research was supported by the European Research Council under the European Union's Seventh Framework Program (FP7/2007-2013) / ERC grant agreement n° [609819], SOMICS. We thank Fanni Takátsy for her help with data collection.”

“This research was supported by the European Research Council under the European Union's Seventh Framework Program (FP7/2007-2013) / ERC grant agreement n° [609819], 961 SOMICS”

6. We noted in your submission details that a portion of your manuscript may have been presented or published elsewhere. [A subset of our data (from Experiments 1 and 2) is published in Dockendorff, M., Schmitz, L., Knoblich, G., & Vesper, C. (2021). Understanding distal goals from proximal communicative actions. Proceedings of the Annual Meeting of the Cognitive Science Society (Vol. 43, No. 43).In Proceedings of the Annual Meeting of the Cognitive Science Society (Vol. 43, No. 43)] Please clarify whether this [conference proceeding or publication] was peer-reviewed and formally published. If this work was previously peer-reviewed and published, in the cover letter please provide the reason that this work does not constitute dual publication and should be included in the current manuscript.

Reviewers' comments:

Reviewer's Responses to Questions

**Comments to the Author**

1. Is the manuscript technically sound, and do the data support the conclusions?

Reviewer #1: Partly

Reviewer #2: Partly

2. Has the statistical analysis been performed appropriately and rigorously? 

Reviewer #1: Yes

Reviewer #2: Yes

3. Have the authors made all data underlying the findings in their manuscript fully available?

Reviewer #1: No

Reviewer #2: Yes

4. Is the manuscript presented in an intelligible fashion and written in standard English?

Reviewer #1: Yes

Reviewer #2: Yes

5. Review Comments to the Author

Reviewer #1: This manuscript presents three experiments that test people’s ability to infer distal goals based on how an agent completes a proximal one. Experiment 1 shows that people associate higher velocities with more distant proximal goals, but that people do not use this “iconic” speed-distance association reliably when predicting a distal goal from a proximal one. Experiment 2 then rules out the possibility that this failure is due to an inability to detect differences in velocities over shorter distances, and Experiment 3 shows that people will use the iconic distance-velocity mapping to infer distal goals after having to infer proximal goals based on velocity.

The manuscript tackles an interesting question using a simple and highly-controlled paradigm, and the data help shed light on people’s ability to infer distal goals from proximal ones. At the same time, the claims and conclusions in the manuscript may not always be well justified by the experimental work, and the manuscript would benefit from a major revision before publication.

Major comments:

1. The manuscript is framed as focusing on communicative action, but it is not clear whether the experiment indeed maps onto communicative expectations. While participants are told that the movements they are watching are communicative, it is not clear whether this affects participant inferences in any way (or whether the modulations in the movement match the types of modulations that people would produce were they trying to communicate). In other words, is it possible that you would obtain identical results in a case where participants do not believe the movements are communicative?

To show that the inferences are truly communicative it would be necessary to run a second condition of Experiment 1 where participants are told that the movement was produced by an agent that completed the task in the absence of observers. Alternatively, if the authors are using a different operationalization of communicative (such as anything that reveals information), it would be helpful if this were explained in the introduction. This point should also ideally be returned to in the discussion (particularly given the introduction’s emphasis on SMC as producing intentional communicative modulations).

2. The experiment, while elegant, might also be fairly transparent to participants given the within-subjects aspect of the design. Is it possible that participants do not hold any general intuitions about the relation between velocity and distal goals, but invoke one here because the task transparently reveals that participants are being tested on how to associate speed with goals? To show that this is not the case, it would be ideal to replicate Experiment 1 and ask participants at the end of the task what they thought the experimenters wanted them to do, and how they decided on their strategy. If this is not possible, then it would be helpful if the discussion acknowledged this potential limitation.

3. Related to this, the manuscript would benefit from more details about the experimental procedure of Study 1.

3.1 First, based on our reading of the familiarization procedure, participants saw the slower normal movement for the near target familiarization and the faster normal movement for the far target familiarization (However, there is some ambiguity in the writing as to whether or not this was actually the case). Is it possible that this familiarization phase biased participants towards a pattern of results that associates faster movements with longer distances (or more generally, could this have highlighted an association between speed and goals, increasing the transparency of the task as discussed above)? The methods section would benefit from additional details about the familiarization design. In particular, it would be helpful if the paper clarified whether participants saw the slower normal movement for the near target and the faster normal movement for the far target). Additionally, the Discussion section should ideally acknowledge the possibility of potential biasing effects on participants’ judgements as a potential limitation.

3.2 Second, it would be helpful if the paper included a more detailed explanation regarding the critical distal goal condition in Study 1. In particular, the “teleporting” aspect of the movement introduced in the familiarization could affect what the participants thought of this goal-directed action, and influence the results. This is currently difficult to evaluate because the explanations about the procedure are a bit too high-level. It would be great if the paper could explain the warm-up and procedure in full detail, including the rationale behind the teleporting component and what the authors intended when designing this condition.

Intuitively, it seemed as though the teleporting aspect was an intentional choice to obscure the relationship between velocity and distance for the distal goal. However, several phrases in the manuscript assume that there was an obvious connection between velocity and distance for the distal goal (e.g., “Taken together, the results of the two experiments raise the possibility that some participants in the Distal goal condition in Experiment 1 might not have been aware of the underlying motor iconic relation between the proximal movement and the distal goal”). Critically, the manuscript raises questions in the Discussion about how participants understood this action sequence, making the choice for teleportation as the second action somewhat unclear. It would be helpful to clarify for readers why the authors chose an unfamiliar second action (i.e., teleportation) for inclusion in this task.

3.3 More generally, the Methods section does not include the directions used to introduce the task to participants, and this makes it difficult to understand how participants might think that the distal goal would shape the proximal goal-directed actions. Were participants given any indication that this was actually how the movement occured (i.e., the demonstrator moved the box to the proximal goal, paused, and then the box teleported itself to the distal goal)? Or were they told that they would not see all of the demonstrator’s movements (i.e. the pausing and subsequent teleporting were alterations to the real movement meant to obscure a more continuous trajectory)?

3.4 Similarly, a more detailed explanation would be helpful when introducing the prediction that participants would have an underlying motor-iconic relation between the proximal movement and the distal goal. Specifically, if participants are not familiar with the type of action used to accomplish a distal goal, why should we predict that they would be able to simulate or guess the “backwards effects” that the distal goal would have on the manner of movement used to achieve the proximal goal? You could imagine a parallel task in which participants are told that they are going to see someone reach for a pencil and then they’ll either dax or blick; and that their task is to predict whether the person will dax or blick. In this hypothetical task, participants would have no familiarity with the distal goal and would be forced to make a best guess based on whatever arbitrary cues are available and therefore it would be inaccurate to say that the participants are interpreting proximal actions in terms of their distal goal. Is it possible that the task in this paper reflects the same process as this thought experiment?

4. Throughout the manuscript, the authors explore the idea that movement modulations might be best thought of as a kind of iconicity in which features of the action represent its goal. However, the relationship between speed and distance might be better conceptualized as indexical, rather than iconic. An indexical sign is caused by what it signifies (e.g., smoke signals fire), whereas an iconic sign resembles what it signifies. Since there is a lawful relationship between the velocity and distance of natural movements, it feels more consistent to view the relationship between distance and velocity as causal, therefore making velocity an indexical sign for distance.

This distinction may be important, because indexical signs are not necessarily communicative (simply doing the action would generate a certain pattern of velocities, which could explain the pattern of results in Study 1’s proximal condition without reference to communication).

The indexical vs. iconic distinction might also be important in relation to the findings in the distal goal condition in Study 1. The unfamiliar movement dynamics (due to the teleportation) in the distal goal could remove the indexical nature of the velocity-distance relationship for participants, inviting truly iconic interpretation. Critically, however, it seems as though either interpretation in the distal condition (fast-far or slow-far) is consistent with an iconic interpretation (slow speed and longer distance doesn’t seem like an impossible connection to draw). It would be helpful if the authors discussed the nature of iconicity in this task more thoroughly and considered whether referring to the relationship as indexical might be more appropriate in some cases.

5. The goal of Study 3 and its implications are a bit difficult to follow. This is in part because the manuscript appears to assume that participants should understand how the distal goal would shape proximal actions (in a manner parallel to the football example provided in the introduction). But it is not clear how participants thought about the teleportation component of the task. To make this interpretation, wouldn’t you need to show that people retain the natural relationship between speed and distance in this task?

Moreover, the indexical vs. iconic distinction may be relevant here. Under the interpretation offered in point 4 above, highlighting indexical relationships in the proximal goal phase of Study 3 could impact subsequent iconic interpretation in the distal goal phase, but arbitrarily choosing the “wrong” iconic interpretation in the distal goal phase would not undermine later indexical interpretation in the proximal goal phase, since the velocity-distance relationship is seen as causal.

Minor comments:

-Ln 35-37: Upon first read, the sentence sounds like physical distance was manipulated (i.e., increasing the separation between distal and proximal). Consider saying instead that adding a distal goal led to more variation in mapping.

-Ln 40-42: The final sentence of the abstract reads as if the paper will include data on modulation production. Consider changing the wording to make it clearer that the paper’s contribution is about how people interpret/use modulations.

-Ln 92-93: The initial definition of distal goals was a bit difficult to follow. While the concept becomes clearer throughout the paper, this preliminary definition could benefit from clarification. In a sense, the proximal goal condition also asks about a goal that goes beyond the observed action since the final goal is occluded and the observed action is only part of the trajectory.

-Ln 214-222: When laying out these predictions, the authors do not mention distal goals specifically. It would be good to make explicit that the authors do not predict any differences between the distal and proximal goal here.

-Ln 298: The data preparation section would be easier to follow if it came after the procedure.

-Ln 321-322: Did participants see the slower normal movement for the near target and the faster normal movement for the far target?

-Study 1 Results: The concepts of mapping and consistency were difficult to follow. Please consider extending the explanation at the beginning of the results section to better explain and contextualize these two measures (e.g., a score of 12 would mean x but a score of -12 would mean y).

-Figures 6 and 7: These figures are difficult to read. Please consider revising the titles of each column.

-It would be helpful to have the warm-up videos and the verbatim task instructions available on OSF.

Reviewer #2: Abstract

Experiments aimed at examining if observers can derive information about distal goals by relying on kinematic modulations of  an actor’s instrumental actions. They designed a paradigm in which participants observed a box moving and had to guess the final location of the box (near vs. Far). Participants were randomly assigned to two distincts conditions (Proximal vs. Distal). Authors manipulated the velocity of the movement of box and expected that this modulation impact the decision of the participants about the final destination of the box.

They conducted three experiment in order to show that 1) participants used movement velocity to choose the final location of the box in both proximal and distal condition (Experiment 1), 2) participants were able to discriminate between “slow” and “fast” movements (Experiment 2), 3) participants used motor-iconic association depending the repetition between “proximal” and “distal” conditions (Experiment 3).

Generally, I think that the paper is well written and the experiments are also well conducted. I have some small concerns about the way authors had framed their research and discuss and interpreted their results.

But I think that my comments below could be address with a minor revision.

Main comments

a. I was wondering why the authors decided to focus only temporal aspect of the movement (i.e. velocity) ? The same team already showed that spatial parameter (i.e., amplitude of the movement is altered during action performing during joint action (e.g., Vesper et al., 2016).

Moreover the design of the experiments allow the authors to test such parameters. It might be interesting to discuss it.

b. Auhors based their prediction on literature on velocity for single movement (e.g., ref 23 in their paper). However, the distal condition is comparable to an action sequence with a stationary position (see 33 l.739). It might be interesting that authors further discuss this part. I recommend that the authors might pay more attention to their data in the distal condition. They reported huge variability for the mapping score for each movement condition. As they also not observed such variability for the consistency score, is it possible that there is different motor-iconic interpretation within their participants ? If it the case how interpret this phenomenon ?

I would suggest that authors pay attention to how participants might infered the weight of the box. For instance, is it possible that participants interpreted the first movement as a resting state before performing the other movement. As is it a sequence small effort for the first movement and then large effort for the second ?

Additional experience (not mandatory)

I was wondering what would participants actually do if they had to move the box in the distal condition. How participants would move an object to the point A ? Both movements will present the same Kinematic ? Authors would assumed that the movement would not be the same ?

Analisys

Authors reported variability in the motor iconic relation for the distal condition. Are the same participants that use an non motor iconic relation across the conditions/trials in the distal condition of the first experiment ?

Minor comments

Experiment 1 - I recommend to switch procedure section before data preparation section

Analysis - I recommend authors to further explain what we could interpreted with their dependant variables.

6. PLOS authors have the option to publish the peer review history of their article (what does this mean?). If published, this will include your full peer review and any attached files.

Reviewer #1: No

Reviewer #2: No

---

## [Author Response · Author response to Decision Letter 0]

27 Oct 2022

Response to Reviewers (responses are indicated by the letter R)

Reviewer #1:

The manuscript tackles an interesting question using a simple and highly-controlled paradigm, and the data help shed light on people’s ability to infer distal goals from proximal ones. At the same time, the claims and conclusions in the manuscript may not always be well justified by the experimental work, and the manuscript would benefit from a major revision before publication.

Major comments:

1. The manuscript is framed as focusing on communicative action, but it is not clear whether the experiment indeed maps onto communicative expectations. While participants are told that the movements they are watching are communicative, it is not clear whether this affects participant inferences in any way (or whether the modulations in the movement match the types of modulations that people would produce were they trying to communicate). In other words, is it possible that you would obtain identical results in a case where participants do not believe the movements are communicative?

To show that the inferences are truly communicative it would be necessary to run a second condition of Experiment 1 where participants are told that the movement was produced by an agent that completed the task in the absence of observers. Alternatively, if the authors are using a different operationalization of communicative (such as anything that reveals information), it would be helpful if this were explained in the introduction. This point should also ideally be returned to in the discussion (particularly given the introduction’s emphasis on SMC as producing intentional communicative modulations).

R: The communicative expectations in our experiments are linked to two aspects: a) our familiarization, in which we explicitly tell participants that the movements are communicative, and b) the exaggerations in the velocity of the movements presented, since these provide useful anticipatory information about the actor’s goals. This latter point is central to the present work as well as to previous studies in sensorimotor communication (SMC), where communication is sometimes taken broadly to mean any kinematic modulation that facilitates an observer’s predictions about an actor’s goals (Pezzulo et al., 2018). We now made this point more explicit in the Introduction (lines 84-93). In so doing, we contrast this broad definition of communication, centering around the role of the receiver in interpreting communicative actions in order to better predict an actor’s goals, to other perspectives on communication.

2. The experiment, while elegant, might also be fairly transparent to participants given the within-subjects aspect of the design. Is it possible that participants do not hold any general intuitions about the relation between velocity and distal goals, but invoke one here because the task transparently reveals that participants are being tested on how to associate speed with goals? To show that this is not the case, it would be ideal to replicate Experiment 1 and ask participants at the end of the task what they thought the experimenters wanted them to do, and how they decided on their strategy. If this is not possible, then it would be helpful if the discussion acknowledged this potential limitation.

R: Our key manipulation of goal type (whether the goal was Proximal or Distal) was a between-subjects manipulation. Moreover, the fact that participants had to associate movements to goals was made explicit to them during the familiarization, where they were told and subsequently shown movements that were unambiguously goal-directed (i.e., “You just saw one of the movements performed by a previous participant. Here, the participant delivered the box to the FAR/NEAR location”). Thus, the goal-directedness of these movements was never concealed from them. (see verbatim task instructions used in the familiarization, now added to OSF, in line 271) However, we don’t think that this is problematic in our study, since the instrumental nature of the movements is essential for participants to understand our stimuli. Crucially, and despite making explicit the goal-directed nature of the movements, we still found variation in the way participants mapped the movements onto their goals in our key condition, the Distal goal condition. This suggests that our task was more than a mere one-to-one mapping exercise between movements and targets, and rather required participants to interpret them with respect to their most likely goals.

We did ask participants a few questions about their overall experience with the task at the end of the experiment. One of the questions was about whether they had a “general strategy to solve the task”, to which the majority replied by describing the particular mapping they used (either motor-iconic or its reversal). We discuss some of these qualitative results (specifically those of the Distal goal condition) in the Discussion of Experiment 1 (lines 494-504).

3. Related to this, the manuscript would benefit from more details about the experimental procedure of Study 1.

3.1 First, based on our reading of the familiarization procedure, participants saw the slower normal movement for the near target familiarization and the faster normal movement for the far target familiarization (However, there is some ambiguity in the writing as to whether or not this was actually the case). Is it possible that this familiarization phase biased participants towards a pattern of results that associates faster movements with longer distances (or more generally, could this have highlighted an association between speed and goals, increasing the transparency of the task as discussed above)? The methods section would benefit from additional details about the familiarization design. In particular, it would be helpful if the paper clarified whether participants saw the slower normal movement for the near target and the faster normal movement for the far target). Additionally, the Discussion section should ideally acknowledge the possibility of potential biasing effects on participants’ judgements as a potential limitation.

R: Across our two between-subject conditions we tried to keep the familiarizations as similar as possible. Specifically, participants were familiarized with two Normal (non-exaggerated) movements, each directed at either the Near or the Far target location. We now made this information more explicit in the description of the Procedure of Experiment 1. (lines 325-337). We also added a file to OSF containing the verbatim task instructions used during the familiarization (see link in line 271)

In the Proximal condition the two Normal movements contained perceivable differences in their velocities (see Figure 2A), and participants may have been biased by such differences already during the familiarization. We now acknowledge this possibility in the Discussion of Experiment 1 (line 457-469). However, we also think that this is a natural consequence of our decision to present real movements during familiarization, rather than artificially modified ones.

3.2 Second, it would be helpful if the paper included a more detailed explanation regarding the critical distal goal condition in Study 1. In particular, the “teleporting” aspect of the movement introduced in the familiarization could affect what the participants thought of this goal-directed action, and influence the results. This is currently difficult to evaluate because the explanations about the procedure are a bit too high-level. It would be great if the paper could explain the warm-up and procedure in full detail, including the rationale behind the teleporting component and what the authors intended when designing this condition. Intuitively, it seemed as though the teleporting aspect was an intentional choice to obscure the relationship between velocity and distance for the distal goal. However, several phrases in the manuscript assume that there was an obvious connection between velocity and distance for the distal goal (e.g., “Taken together, the results of the two experiments raise the possibility that some participants in the Distal goal condition in Experiment 1 might not have been aware of the underlying motor iconic relation between the proximal movement and the distal goal”). Critically, the manuscript raises questions in the Discussion about how participants understood this action sequence, making the choice for teleportation as the second action somewhat unclear. It would be helpful to clarify for readers why the authors chose an unfamiliar second action (i.e., teleportation) for inclusion in this task.

R: We now clarify the rationale behind the teleporting component in the Present study section (lines 231-238) There are two main reasons why the Distal goal condition included a “teleportation”. First, we wanted to have an explicit separation between the movements and the target locations. Second, we wanted to keep, across our two between-subject conditions, a single sliding movement that participants would need to use to infer the proximal or distal goal. In order to achieve these two requirements, we decided to implement a “teleportation” component in the Distal goal condition, that is, a clear separation between the observed movement and the goal. In sum, the purpose of the teleportation was not to obscure the relationship between movements and goals, but rather to have a salient spatial and temporal separation between these two.

Furthermore, we hypothesized that the underlying relation between movement velocity and distance in the Distal goal condition is grounded on previous findings showing that unconstrained aiming movements tend to be performed with higher velocities when the targets are further away (Jeannerod, 1984). When this lawful relation is invoked by an observer during action observation in order to predict a distal goal, then it can be considered motor-iconic (lines 191-199). One of the main purposes of our study was precisely to answer the empirical question of whether such a relation would be preserved in the Distal goal condition.

Finally, we acknowledge, in the General Discussion, that this explicit separation may have had an effect on participant’s responses. Specifically we mention the possibility that some participants may have taken the “teleportation” of the box as an invisible implied second movement, which they had to imagine or simulate once the box suddenly disappeared from the display (line 819-827).

3.3 More generally, the Methods section does not include the directions used to introduce the task to participants, and this makes it difficult to understand how participants might think that the distal goal would shape the proximal goal-directed actions. Were participants given any indication that this was actually how the movement occured (i.e., the demonstrator moved the box to the proximal goal, paused, and then the box teleported itself to the distal goal)? Or were they told that they would not see all of the demonstrator’s movements (i.e. the pausing and subsequent teleporting were alterations to the real movement meant to obscure a more continuous trajectory)?

R: We have now added a more detailed description of the familiarization in the Procedure of Experiment 1 (lines 325-337) and added a file to the OSF repository (link in line 271) containing the exact wording used in both familiarizations. With respect to the teleportation, we simply told participants in the Distal goal condition that they would “see different animations of the box below moving along the black line to the grey area in the middle of the screen.” Next, we added that “the box will then be delivered to one of the two green locations you see on the right.” We didn’t provide any further information about the “teleportation” nor its relation to the first movement. This was done on purpose, precisely because we wanted to see how participants would understand such separation without biasing them to speculate about whether the actor played a role in triggering the teleportation.

3.4 Similarly, a more detailed explanation would be helpful when introducing the prediction that participants would have an underlying motor-iconic relation between the proximal movement and the distal goal. Specifically, if participants are not familiar with the type of action used to accomplish a distal goal, why should we predict that they would be able to simulate or guess the “backwards effects” that the distal goal would have on the manner of movement used to achieve the proximal goal? You could imagine a parallel task in which participants are told that they are going to see someone reach for a pencil and then they’ll either dax or blick; and that their task is to predict whether the person will dax or blick. In this hypothetical task, participants would have no familiarity with the distal goal and would be forced to make a best guess based on whatever arbitrary cues are available and therefore it would be inaccurate to say that the participants are interpreting proximal actions in terms of their distal goal. Is it possible that the task in this paper reflects the same process as this thought experiment?

R: We have now spelled out the predictions for the Distal goal condition in more detail in the section on Motor iconicity in SMC (lines 191-203). We also specify why we expect to find more motor-iconic mappings in this condition. Furthermore, we believe that the analogy with the thought experiment proposed by R1 applies only to a certain extent. First, our task presents participants with signals (i.e., movements) and referents (i.e., goals) which are both part of a single broad domain, that of human actions. Relatedly, we predicted that participants would be able to identify the underlying motor-iconic relation between the movements and their goals, even when these goals are distal, based on previous findings from the motor control literature. Thus, the relationship between movements and goals is not totally arbitrary, as it would be the case if the task involved dax and blicks.

4. Throughout the manuscript, the authors explore the idea that movement modulations might be best thought of as a kind of iconicity in which features of the action represent its goal. However, the relationship between speed and distance might be better conceptualized as indexical, rather than iconic. An indexical sign is caused by what it signifies (e.g., smoke signals fire), whereas an iconic sign resembles what it signifies. Since there is a lawful relationship between the velocity and distance of natural movements, it feels more consistent to view the relationship between distance and velocity as causal, therefore making velocity an indexical sign for distance. This distinction may be important, because indexical signs are not necessarily communicative (simply doing the action would generate a certain pattern of velocities, which could explain the pattern of results in Study 1’s proximal condition without reference to communication).

The indexical vs. iconic distinction might also be important in relation to the findings in the distal goal condition in Study 1. The unfamiliar movement dynamics (due to the teleportation) in the distal goal could remove the indexical nature of the velocity-distance relationship for participants, inviting truly iconic interpretation. Critically, however, it seems as though either interpretation in the distal condition (fast-far or slow-far) is consistent with an iconic interpretation (slow speed and longer distance doesn’t seem like an impossible connection to draw). It would be helpful if the authors discussed the nature of iconicity in this task more thoroughly and considered whether referring to the relationship as indexical might be more appropriate in some cases.

R: We now make more explicit how motor-iconicity can be understood in the case of the Distal goal condition in the section on Motor-iconicity in SMC (lines 191-203), and concede the point that indexicality might be better suited to describe the results of the Proximal goal condition in the General Discussion (lines 789-818). In this point, we fully agree with the comments made by Reviewer 1 and we thank him/her for elaborating on this idea. Specifically, we argue that the lawful relationship between velocity and goals might be seen as one in which observers simply need to extract some causal or spatiotemporal (and therefore indexical) regularity between the velocity of the movements they observe and their most likely proximal goals.

5. The goal of Study 3 and its implications are a bit difficult to follow. This is in part because the manuscript appears to assume that participants should understand how the distal goal would shape proximal actions (in a manner parallel to the football example provided in the introduction). But it is not clear how participants thought about the teleportation component of the task. To make this interpretation, wouldn’t you need to show that people retain the natural relationship between speed and distance in this task?

Moreover, the indexical vs. iconic distinction may be relevant here. Under the interpretation offered in point 4 above, highlighting indexical relationships in the proximal goal phase of Study 3 could impact subsequent iconic interpretation in the distal goal phase, but arbitrarily choosing the “wrong” iconic interpretation in the distal goal phase would not undermine later indexical interpretation in the proximal goal phase, since the velocity-distance relationship is seen as causal.

R: We have clarified in the manuscript what we mean by motor-iconic relation and the reasons for why such relation can also be applied to the Distal goal condition (lines 191-203)

We now included, in the General discussion, the idea that deriving goals in the Proximal goal condition is in some sense analogous to the process of interpreting indexical signs. We argue this on the basis of the arguments made by the reviewer above, where an indexical relation is defined in terms of the causal relation holding between the signal and the referent (lines 789-818). We also discuss the possibility that such causal, indexical relations, might provide an explanation for the precedence effect found in Experiment 3, where the Proximal goal trials were found to affect subsequent Distal goal trials (lines 814-818)

Minor comments:

-Ln 35-37: Upon first read, the sentence sounds like physical distance was manipulated (i.e., increasing the separation between distal and proximal). Consider saying instead that adding a distal goal led to more variation in mapping. 

R: We changed it to “Adding a distal goal” (line 35)

-Ln 40-42: The final sentence of the abstract reads as if the paper will include data on modulation production. Consider changing the wording to make it clearer that the paper’s contribution is about how people interpret/use modulations. 

R: We changed it to “can be used to infer more distal goals” (line 42)

-Ln 92-93: The initial definition of distal goals was a bit difficult to follow. While the concept becomes clearer throughout the paper, this preliminary definition could benefit from clarification. In a sense, the proximal goal condition also asks about a goal that goes beyond the observed action since the final goal is occluded and the observed action is only part of the trajectory. 

R: We have clarified the preliminary definition, now it reads: goals that go beyond the observed action and thus are only attained after the achievement of a more proximal (sub-) goal first. (lines 101-102)

-Ln 214-222: When laying out these predictions, the authors do not mention distal goals specifically. It would be good to make explicit that the authors do not predict any differences between the distal and proximal goal here.  

R: We did predict motor-iconic mappings for the Distal goal condition. This is made more explicit now (lines 191-203 and 244-249)

-Ln 298: The data preparation section would be easier to follow if it came after the procedure. 

R: We moved this section to the beginning of the Results section (now lines 358-376)

-Ln 321-322: Did participants see the slower normal movement for the near target and the faster normal movement for the far target? 

R: Yes, and although we do not mention it explicitly in the Procedure, we do acknowledge this point in the Discussion of Experiment 1, specifically with respect to the Proximal goal condition. In the Distal goal condition the two Normal movements were practically identical. (lines 325-337)

-Study 1 Results: The concepts of mapping and consistency were difficult to follow. Please consider extending the explanation at the beginning of the results section to better explain and contextualize these two measures (e.g., a score of 12 would mean x but a score of -12 would mean y). 

R: We moved the “Data preparation” sub-section to the beginning of the Results section. We included an explanation of what a Consistency score of 0 and 12 mean, and what a Mapping score of +12, 0 and -12 mean (lines 358-376)

-Figures 6 and 7: These figures are difficult to read. Please consider revising the titles of each column. 

R: We changed the titles of these two figures.

-It would be helpful to have the warm-up videos and the verbatim task instructions available on OSF.

R: We added the file to OSF (see link in line 271)

Reviewer #2:

a. I was wondering why the authors decided to focus only temporal aspect of the movement (i.e. velocity) ? The same team already showed that spatial parameter (i.e., amplitude of the movement is altered during action performing during joint action (e.g., Vesper et al., 2016). Moreover the design of the experiments allow the authors to test such parameters. It might be interesting to discuss it.

R: Our task builds on Vesper et al.’s previous work, but we decided to restrict our study to the manipulation of a single movement parameter, namely peak velocity. We discuss the possibility in the General discussion (in line 828-846) that communicators would try to convey information about proximal and distal goals using other movement parameters, such as movement duration or changes in movement direction. Furthermore, we describe the communicative strategies that participants reported they would use, were they asked to use the box to communicate about the target locations (lines 839-846)

b. Auhors based their prediction on literature on velocity for single movement (e.g., ref 23 in their paper). However, the distal condition is comparable to an action sequence with a stationary position (see 33 l.739). It might be interesting that authors further discuss this part. I recommend that the authors might pay more attention to their data in the distal condition. They reported huge variability for the mapping score for each movement condition. As they also not observed such variability for the consistency score, is it possible that there is different motor-iconic interpretation within their participants ? If it the case how interpret this phenomenon ?

R: We discuss some of the reasons why we think that there is higher variability in the mappings of the Distal goal condition in the General discussion (line 764-788). Specifically, we point to the possibility that the Distal goal condition, because of its separation between the movement and the goal, might be inducing a more “symbolic” form of interpretation, analogous to the way we sometimes interpret manual gestures. We also allude to the possibility (lines 819-827) that the movement presented in the Distal goal condition may have been interpreted as the first movement component of a two-step action sequence. The absence of a second movement component, and its replacement for a sudden “teleportation”, may have also led to higher variability in that condition.

I would suggest that authors pay attention to how participants might infered the weight of the box. For instance, is it possible that participants interpreted the first movement as a resting state before performing the other movement. As is it a sequence small effort for the first movement and then large effort for the second ?

R: We presented participants with smooth sliding movements with low-friction in order to give the impression that no big effort was required to move the box. We also added a black line along which the box was moved and a cursor on top of the box so as to induce in participants the impression that these were easy-to-perform movements. We now discuss in the General discussion the role that objects, with their specific physical properties (e.g, weight, size), might play in the way communicators use communicative modulations (lines 847-858)

I was wondering what would participants actually do if they had to move the box in the distal condition. How participants would move an object to the point A ? Both movements will present the same Kinematic ? Authors would assumed that the movement would not be the same ?

R: We did ask participants, once they completed the task, how they would try to communicate to someone else about the green locations if they were only allowed to use the movements of the box. We now added in the General Discussion (lines 839-846) a few examples which highlight the variability of their strategies, were they to use the box to communicate about the target locations.

Authors reported variability in the motor iconic relation for the distal condition. Are the same participants that use an non motor iconic relation across the conditions/trials in the distal condition of the first experiment ?

R: Our two goal conditions (Proximal and Distal) were manipulated as between-subject variables in Experiment 1. Those who use a non-motor-iconic mapping do so consistently, as indicated by the presence of participants in the Distal goal condition whose Mapping Score is close to -12. 

Minor comments

Experiment 1 - I recommend to switch procedure section before data preparation section. 

R: We moved the Data Preparation section to the beginning of the Results section (lines 358-376)

Analysis - I recommend authors to further explain what we could interpreted with their dependant variables. R: The Data Preparation section of Experiment 1 includes a description of what the different values in the Consistency and Mapping scores mean. For example, a Consistency score of 12 means that the participant was producing consistent velocity-to-location mappings throughout the experiment, but it does not tell us in which direction (motor-iconic or its reversal). That is precisely what the Mapping score is for. A Mapping score of +12 specifies that the mapping is fully motor-iconic, whereas a Mapping score of -12 means that the participant reversed the motor-iconic mapping altogether (lines 366-376). In our analyses, we compare Consistency and Mapping scores to a baseline (i.e., 0), which indicates either a non-consistent baseline or a lack of mapping preference (motor-iconic or its reversal). This is also described in the Results section of Experiment 1 (lines 378-380 and 411-413)

---

## [Decision Letter · Decision Letter 1]

26 Dec 2022

Understanding others’ distal goals from proximal communicative actions

PONE-D-22-13033R1

Dear Dr. Dockendorff,

I am pleased to inform you that your manuscript has been judged scientifically suitable for publication and will be formally accepted for publication once it meets all outstanding technical requirements.

With kind regards,

Cédric A. Bouquet

Academic Editor

PLOS ONE

Reviewers' comments:

Reviewer's Responses to Questions

**Comments to the Author**

1. If the authors have adequately addressed your comments raised in a previous round of review and you feel that this manuscript is now acceptable for publication, you may indicate that here to bypass the “Comments to the Author” section, enter your conflict of interest statement in the “Confidential to Editor” section, and submit your "Accept" recommendation.

Reviewer #1: All comments have been addressed

Reviewer #2: All comments have been addressed

2. Is the manuscript technically sound, and do the data support the conclusions?

Reviewer #1: Yes

Reviewer #2: Yes

3. Has the statistical analysis been performed appropriately and rigorously? 

Reviewer #1: Yes

Reviewer #2: Yes

4. Have the authors made all data underlying the findings in their manuscript fully available?

Reviewer #1: Yes

Reviewer #2: Yes

5. Is the manuscript presented in an intelligible fashion and written in standard English?

Reviewer #1: Yes

Reviewer #2: Yes

6. Review Comments to the Author

Reviewer #1: The revision addresses my previous concerns. I appreciate the authors’ careful inclusion of additional explanations and caveats as requested and I recommend accepting the article for publication.

Reviewer #2: I think that the authors did a great job. I don't have any major comments. I think that the paper is ready to go.

Best regards

7. PLOS authors have the option to publish the peer review history of their article (what does this mean?). If published, this will include your full peer review and any attached files.

Reviewer #1: No

Reviewer #2: No

---

## [Editor Report · Acceptance letter]

2 Jan 2023

PONE-D-22-13033R1 

Understanding others’ distal goals from proximal communicative actions 

Dear Dr. Dockendorff:

I'm pleased to inform you that your manuscript has been deemed suitable for publication in PLOS ONE. Congratulations! Your manuscript is now with our production department. 

Kind regards, 

on behalf of

Dr. Cédric A. Bouquet 

Academic Editor

PLOS ONE